# Correlated states in doubly-aligned hBN/graphene/hBN heterostructures

Xingdan Sun [1,2,14], Shihao Zhang [3,4,14], Zhiyong Liu[1,2,14], Honglei Zhu[1,2], Jinqiang Huang[1,2], Kai Yuan[5,6], Zhenhua Wang[1,2 ✉], Kenji Watanabe [7], Takashi Taniguchi [8], Xiaoxi Li[1,2,9,10], Mengjian Zhu[11], Jinhai Mao [12], Teng Yang [1,2], Jun Kang[13 ✉], Jianpeng Liu [3,4 ✉], Yu Ye [5,6 ✉], Zheng Vitto Han[9,10 ✉] & Zhidong Zhang [1,2]

Interfacial moiré superlattices in van der Waals vertical assemblies effectively reconstruct the crystal symmetry, leading to opportunities for investigating exotic quantum states. Notably, a two-dimensional nanosheet has top and bottom open surfaces, allowing the specific case of doubly aligned super-moiré lattice to serve as a toy model for studying the tunable lattice symmetry and the complexity of related electronic structures. Here, we show that by doubly aligning a graphene monolayer to both top and bottom encapsulating hexagonal boron nitride (h-BN), multiple conductivity minima are observed away from the main Dirac point, which are sensitively tunable with respect to the small twist angles. Moreover, our experimental evidences together with theoretical calculations suggest correlated insulating states at integer fillings of −5, −6, −7 electrons per moiré unit cell, possibly due to inter-valley coherence. Our results provide a way to construct intriguing correlations in 2D electronic systems in the weak interaction regime.

[1] Shenyang National Laboratory for Materials Science, Institute of Metal Research, Chinese Academy of Sciences, 110016 Shenyang, China. [2] School of Material Science and Engineering, University of Science and Technology of China, 230026 Anhui, China. [3] School of Physical Science and Technology, ShanghaiTech University, 200031 Shanghai, China. [4] ShanghaiTech Laboratory for Topological Physics, ShanghaiTech University, 200031 Shanghai, China. [5] State Key Lab for Mesoscopic Physics, Nano-optoelectronics Frontier Center of the Ministry of Education, School of Physics, Peking University, 100871 Beijing, China. [6] Collaborative Innovation Center of Quantum Matter, 100871 Beijing, China. [7] Research Center for Functional Materials, National Institute for Materials Science, 1-1 Namiki, Tsukuba 305-0044, Japan. [8] International Center for Materials Nanoarchitectonics, National Institute for Materials Science, 1-1 Namiki, Tsukuba 305-0044, Japan. [9] State Key Laboratory of Quantum Optics and Quantum Optics Devices, Institute of Opto-Electronics, Shanxi University, 030006 Taiyuan, China. [10] Collaborative Innovation Center of Extreme Optics, Shanxi University, 030006 Taiyuan, China. [11] College of Advanced Interdisciplinary Studies, National University of Defense Technology, 410073 Changsha, China. [12] School of Physical Sciences and CAS Center for Excellence in Topological Quantum Computation, University of Chinese Academy of Sciences, Beijing, China. [13] Beijing Computational Science Research Center, 100193 Beijing, China. [14] These authors contributed equally: Xingdan Sun, Shihao Zhang, Zhiyong Liu. ✉email: zhwang@imr.ac.cn; jkang@csrc.ac.cn; liujp@shanghaitech.edu.cn; ye_yu@pku.edu.cn; vitto.han@gmail.com

L attice symmetry of a crystal often defines its fundamental physical properties, yet the measures to manipulate in real space the crystallographic symmetries are so far rather limited in solid-state systems. Owing to the feasibility of controllable twisting of van der Waals (vdW) two-dimensional (2D) building blocks, a direct reconstruction of the interfacial moiré superlattice can be realized[1,2], leading to the modifications of electronic band structure of such hybrid systems. As a result, emerging quantum phenomena, such as strong-correlation superconductivity,[3–5] orbital ferromagnetism,[6–8] unusual excitonic properties,[9,10] and nematic electronic orderings,[11–14] have been intensively investigated in twist-angled 2D heterostructures that are fundamentally influenced by the moiré superpotentials.

Although moiré-dominated electronic and optical behaviours have been largely investigated, it is noticed that a 2D vdW electronic layer with duplicated moiré superlattices on both of its surfaces is of particular interests. However, restricted by a sophisticated stacking and twisting operation of doubly aligned moiré superlattice, very few reports have been focused on such a model.[15–18] Indeed, doubly aligned h-BN/graphene/h-BN interfaces give rise to an extra moiré wavelength (defined as composite super-moiré) that is possible to arrange the spectrum reconstruction at arbitrary low energies.[17] More recently, scanning probe tip-assisted dynamic rotating of a top h-BN reveals a subtle distinction of symmetry for 0°/0° and 0°/60° configurations of the h-BN/graphene/h-BN doubly aligned sandwich.[15] Nevertheless, experimental demonstrations of such model systems in a broader gate range and more detailed understandings of the unusual band modification, such as possible flat bands and correlated electronic states, remain unexplored.

In this work, we investigate the case of doubly aligned h-BN/graphene/h-BN devices, in which close-to-zero angle alignments were achieved on both top and bottom graphene/h-BN interfaces, allowing one to fully map the detailed fillings of each emerging minibands thanks to the composite super-moiré lattices. We first demonstrate theoretically that in the scenario when encapsulating h-BNs are in perfect alignment (i.e., 0°–0°–0° aligned), there is only one single moiré superlattice, which yields expected flat minibands. Experimentally, the minibands of doubly aligned composite moiré in h-BN/graphene/h-BN heterostructure can be sensitively tuned by the twisting angle. In the best-aligned double moiré sample S7, we observed resistive peaks at fillings of $-4$, $-8$, $-12$ electrons per moiré unit cell. In other samples S12 and S4 with slightly different, yet rather close to zero, dual twists from top and bottom h-BN/graphene interfaces, evidence of possible correlated insulating (CI) states was observed in $-5$, $-6$, $-7$ electrons per moiré unit cell. Our results suggest that lattice reconstruction of graphene using dually aligned moiré superpotential can be an effective way to realize correlated states.

## Results

**Characterizations of doubly aligned heterostructures**. Monolayered graphene and thin encapsulating h-BN flakes were exfoliated from bulk crystals and stacked with the dry-transfer method.[19] The graphene/h-BN vertical heterostructure was patterned into Hall bars with electrodes edge-contacted. The device was equipped with Au top and graphite bottom gates, as sketched in Fig. 1a (See also Supplementary Fig. 1). A zoomed-in art view is illustrated in Fig. 1b, with the small rotation angle between the top and bottom graphene/h-BN interfaces denoted as $\theta_t$ and $\theta_b$, respectively. Experimentally, the alignment between graphene and the two encapsulating h-BN flakes was realized by carefully aligning the exfoliated straight edges of each flake using the optical micrograph images with a graphic analysis tool software. An example of optically well-aligned sample is given in Fig. 1c,

with a typical dual-gated device using this doubly aligning method shown in Fig. 1d.

It is rather evident that when the three layers are in the alignment regime, the first hole-side satellite resistive peak $R_{-n_s}$ ($n_s$ corresponding to a carrier filling of $4n_0 = \frac{8}{\sqrt{3}L^2}$, with $n_0$ defined as one electron filling per moiré unit cell area $A = \frac{\sqrt{3}L^2}{2}$. $L$ is the moiré wavelength[15]) will be about 1.5 times higher than the central major Dirac peak (with a resistance of $R_D^{neutral}$) at the charge neutrality at room temperature.[15] Such a "well-aligned regime" has a tolerance of less than 1°.[15] Therefore, the above feature of field-effect curves can be a criterion to distinguish devices with different alignments. Indeed, as shown in Fig. 1e, typical samples with different $\theta_t/\theta_b$ values obtained by measuring their straight edges in the optical images exhibit significant variations in their room temperature field effect curves. As a result, sample S8 has a best alignment of $\theta_t/\theta_b$, yielding a field effect curve with $R_{-n_s}$ higher than $R_D^{neutral}$. Meanwhile, other samples with larger mis-alignment showing lower $R_{-n_s}$ than $R_D^{neutral}$, with the satellite peak disappearing in, for example, sample S20. Raman spectroscopy is another means to check the twist angles of the doubly aligned h-BN/graphene/h-BN. As indicated in Fig. 1f, the corresponding devices in Fig. 1e show distinct Raman spectra. The best-aligned sample yields a graphene 2D Raman peak with slightly wider full-width at the half maximum (FWHM) and lowest peak height compared to other less-aligned samples, in agreement with previously reported results.[15,17]

**Band structure of doubly aligned h-BN/graphene/h-BN**. Before further experimental measurements, we study the electronic band structures of the doubly aligned h-BN/graphene/h-BN with a composite moiré superlattice (tight-binding simulations and continuum models are both used, and found to be consistent. See "Methods" and Supplementary Note 1). In the case of monolayer graphene/h-BN single moiré superlattice, the original band structure of graphene will be folded and give rise to, away from the major Dirac cone, duplicates of a set of mini Dirac cones at electron filling of $\pm n_s$.[1,2] The composite moiré superlattice with perfect alignment 0°–0°–0° (illustrated in Fig. 2a) however splits the hole-sided bands at higher energy into minibands with fully developed gaps ($\Delta_i$, $i = 1, 2, 3$) at electron fillings of $-4 n_0$, $-8 n_0$ and $-12 n_0$, as illustrated in Fig. 2b. Compared to the single interfaced h-BN/graphene heterostructure, the consequence of a doubly aligned h-BN/graphene/h-BN moiré superlattice is the strong band anti-crossing originated from an extra moiré potential in the Hamiltonian (Supplementary Note 1). For the specific scenario of perfect alignment, we calculated via continuum model the bandwidth of the first three bands below the charge neutrality, shown in Fig. 2c. It is seen that band I and II increase monotonously with twist angle $\theta$, while band III has a minimum at about 0.65° with a bandwidth of about 35 meV. More details can be found in the Supplementary Information.

**Transport measurements of doubly aligned h-BN/graphene/h-BN**. Bearing in mind with a band structure of 0°–0°–0° illustrated in Fig. 2b, we now come to the experimental examination of the theoretical models. A typical sample S7 was characterized by recording field-effect curves as a function of magnetic field at 45 mK. It develops into a Landau fan with a fractal structure known as the Hofstadter butterfly, which can be renormalised into a diagram defined by the Diophantine equation[20–22]:

$$\frac{n}{n_0} = \nu \frac{\phi}{\phi_0} + s \qquad (1)$$

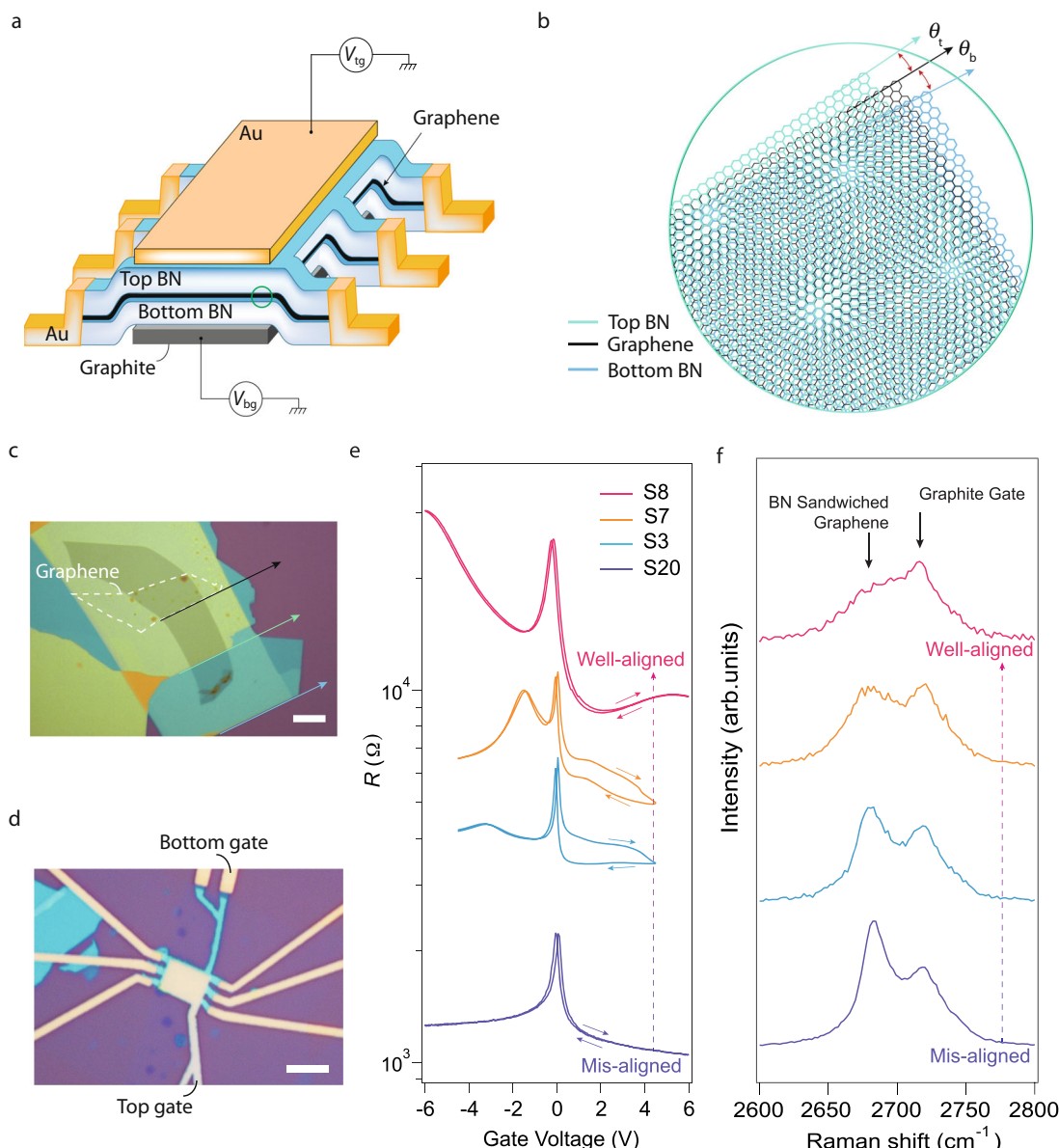

**Fig. 1 Doubly aligned h-BN/graphene/h-BN. a** Schematics of a device with dual moiré, equipped with dual gates. $V_{tg}$ and $V_{bg}$ represent voltages of the top and bottom gate, respectively. **b** Art view of the lattice configuration with twist angles ($\theta_t$ and $\theta_b$) between graphene and the top/bottom h-BN, as marked in the circled area in (**a**). **c** Optical micrograph image of the raw flakes stacked with their exfoliated straight edges (marked as coloured arrow) aligned. The white dashed line shows the contour of graphene. **d** Optical micrograph image of a typical completed device, consisting of a dual-gated doubly aligned h-BN/graphene/h-BN heterostructure. Scale bars in (**c**) and (**d**) are 10 $\mu$m and 5 $\mu$m, respectively. **e** Field effect curves tested in ambient condition of several typical devices, with different twist angles of $\theta_t$ and $\theta_b$. Trace and re-trace curves recorded by sweeping up and down the gate voltages, as indicated by the arrows. **f** Raman spectra in the vicinity of graphene 2D peak of the corresponding devices in (**e**), and illustrated using the same colour code as in (**e**), with the dashed line arrows indicating the transition from misaligned to aligned regime. Scales are renormalized according to the peak height of the graphite gate and shifted for clarity.

where topological integers $\nu$ denotes the Hall conductivity in units of a conductance quantum e²/h, and $s$ is the index of band filling. $\phi = B \cdot A$ is flux per moiré unit area at magnetic field $B$, and $\phi_0 = h/e$ is a flux quantum with $h$ being the Planck's constant. It is seen in Fig. 2d that pseudogaps at each filling of $-4\,n_0$, $-8\,n_0$, $-12\,n_0$ can be extrapolated from the Landau fans at $B = 0$, agreeing well with the expected minibands filling for perfect doubly aligned h-BN/graphene/h-BN. Typical subsets of the quantum Hall states (QHS) that satisfy Eq. (1) are identified as black solid lines in Fig. 2e, with topological index $\nu$ to be ±2, ±6, ±10, etc. More data on other samples with less well alignment can be found in Supplementary Figs. 2–9.

Except for those resistive maxima at full band fillings, extra resistive peaks can be observed. These peaks may come from the slightly non-perfect alignment. To quantitatively analyse them, a phenomenological model was given, which, for simplicity, assumes that the $-4\,n_0$ filling corresponds a fixed h-BN/graphene moiré with a smaller twist (wavelength $L_\alpha$), and all extra peaks are from another moiré with a larger twist (wavelength $L_\beta$). Thus, one can then extract and fix $L_\alpha$, and solve $L_\beta$ using the formula[17,23]

$$l_{\alpha,\beta} = \frac{4\pi}{\sqrt{3}a}\sqrt{\delta^2 + (\theta^{\alpha,\beta})^2} \qquad (2)$$

where $a$ is graphene's lattice constant, $l_\beta$ is the reciprocal lattice

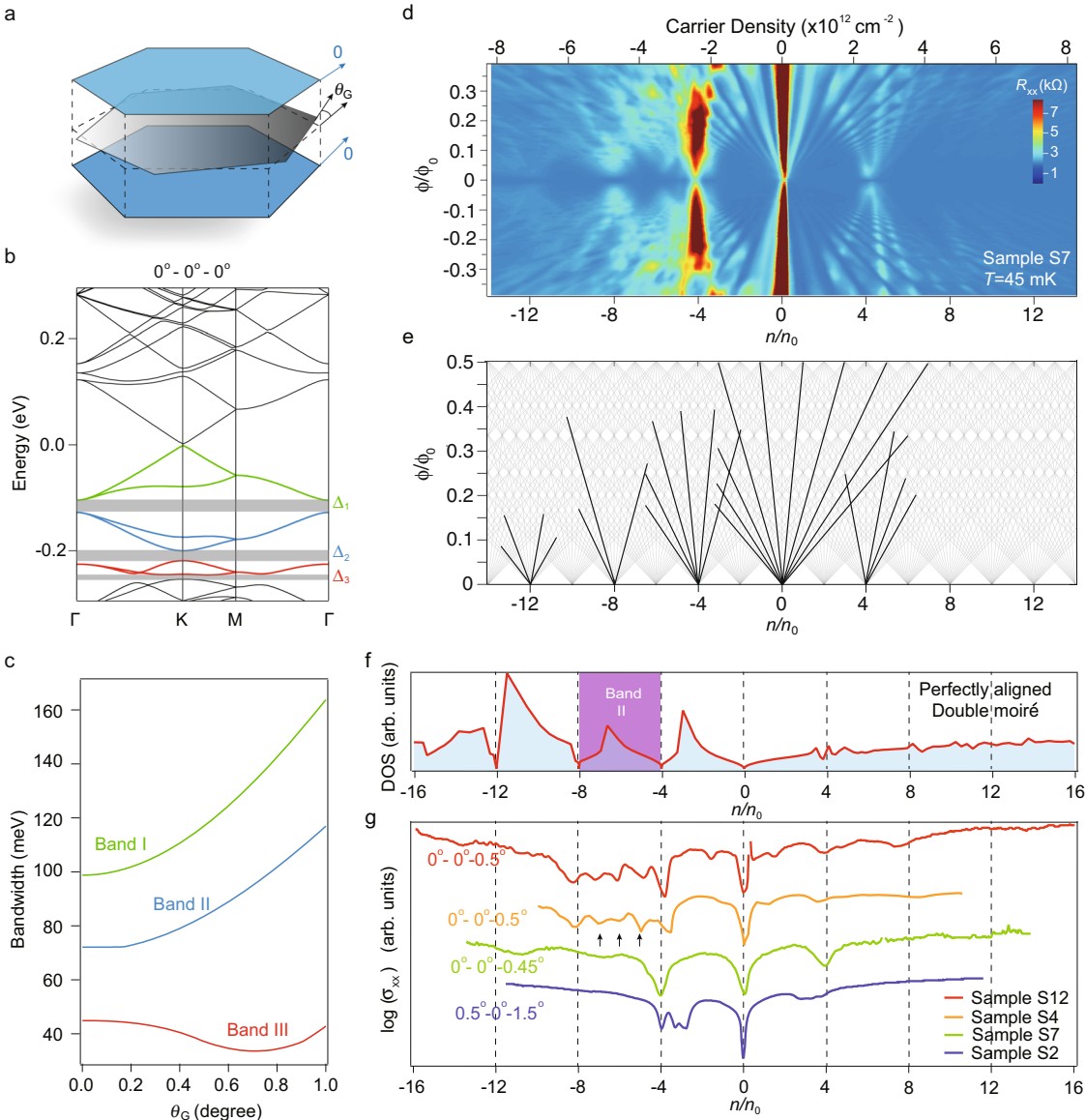

**Fig. 2 Band fillings of doubly aligned h-BN/graphene/h-BN superlattice. a** Schematics of h-BN/graphene/h-BN heterostructure with top and bottom h-BN (blue layer) perfectly aligned and the graphene (grey layer) slightly rotated at $\theta_G$, with its band structure calculated in (**b**). **c** Illustration of bandwidth of the corresponding gaps in (**b**) as a function of $\theta_G$. **d** Landau fan of sample S7 plotted in the space of $\phi/\phi_0$ versus $n/n_0$. $\phi/\phi_0$ and $n/n_0$ are the normalized magnetic flux and carrier density (see Eq. (1)), respectively. **e** Landau fan renormalised into a diagram defined by the Diophantine equation. Solid black lines are selected major Landau levels in (**d**). **f** Density of states (DOS) calculated from band structure in (**b**). **g** Field effect curves measured from different samples at $B = 0$ and $T = 45$ mK, with their twist angles obtained using Eq. (2). Solid black arrows in (**g**) indicate the correlated insulating states at integer fillings of $-5$, $-6$ and $-7$ $n_0$. Vertical dashed lines in (**f**) and (**g**) correspond to band fillings.

vector of $L_\beta$, and $\delta$ is the lattice mismatch between graphene and h-BN[24]. Experimentally, $L_\alpha$ can be larger than the theoretical maximum of 13.8 nm in a graphene/h-BN superlattice, due to possibly existing strains in the heterostructures[17,25]. The above-mentioned analytical formula yields six solutions at each side of the charge neutrality, because of the six vectors formed by the two of moiré Brillouin zone from top and bottom surfaces of graphene[17,23]. With this method, one can have access to the relatively precise twist angles in measured samples via their resistive peaks in the field-effect spectra. For example, twist angles in sample S7 is found to be of 0°–0°–0.45°. Besides the above discussed geometrical model, we also performed atomic force microscopy characterizations of the doubly aligned moiré super-lattice in real space, as shown in Supplementary Figs. 10, 11.

## Discussion

With sample S7, we obtained a filling sequence that resembles perfect aligned 0°–0°–0° moiré superlattice. The minibands are actually super sensitive to twisting angles of each constituent layer. When at slightly different finite dual twists, the resulted composite moiré superlattices will lead to tunable resistive peaks in their spectra of field effect. Calculated density of states (DOS) for the band structure in Fig. 2b is plotted in Fig. 2f, while Fig. 2g summarizes typical samples with different twist angles. By quantifying their exact alignment angles using the same method for sample S7 (more details can be found in Supplementary Figs. 6, 9), we can thus overlay their field-effect curves at $B = 0$ in the renormalized axis of $n/n_0$, in Fig. 2g. It is seen that, even with a tiny difference from 0°–0°–0.45° (sample S7) to 0°–0°–0.5°

(sample S12 and S4), the resistive peaks are largely tuned. For sample S2 with relatively larger twist angle (0.5°–0°–1.5°), only $n = 4\ n_0$ band filling was observed. Notice that samples S12 and S4 are almost identical in their field-effect curves. Interestingly, resistive peaks in the second miniband are seen in them, as indicated by the black solid arrows in Fig. 2g. It is seen that those resistive peaks correspond to integer fillings of $-5\ n_0$, $-6\ n_0$ and $-7\ n_0$, where the host miniband has calculated bandwidth to be of a few tens of meV, marked in purple. It thus suggests possible CI behaviour as observed in other flat band samples previously reported, depicted by the correlation-driven gap opening picture at integer fillings of the moiré unit cell.[4,26–28] Notably, flat band with bandwidth of similar energy was reported theoretically,[29] and were found experimentally possible for the construction of correlated superconductivity in Bernal-stacking bilayer graphene/h-BN superlattice[30]. Among the over 60 samples fabricated, we have 5 dual-moiré samples showing the correlated states at integer fillings from $-5$ to $-7\ n_0$. However, none of them exhibit superconductivity in a reachable doping range and to the lowest temperature we could obtain. Possible explanations for the absence of superconductivity in our samples, even though correlated states signatures are observed, could be that the microscopic mechanism of interaction-induced superconductivity in such as magic angle twisted bilayer graphene are different from that in our case.

To further characterize the CI states in such as sample S12, we performed temperature dependence of its field-effect curves, shown as the colour map in Fig. 3a. It is observed that at temperatures at the order of 100 K, multiple peaks start to develop, with the line cuts of field-effect curves at different temperatures shown in Fig. 3b. Some of the resistive peak at each side of the charge neutrality develops into fine structures, that can be identified as either band fillings (marked by black solid triangles), or composite moiré peaks (marked by black solid circles)(more details of other samples can be seen in Supplementary Figs. 12–16). Notably, the broad satellite peak on the hole-side gradually deconvolutes into individual resistive peaks at $-4$, $-5$, $-6$, $-7$ and $-8\ n_0$ at the ground state. These insulating pseudogaps are located in band II, with the ones at fillings of $-5$, $-6$ and $-7\ n_0$ attributed to electron correlation, as in the non-interacting picture, the doped system is metallic at these three fillings. We thus extract the thermal activation gaps for representative peaks marked by black dashed lines in Fig. 3b, and fitted with the Arrhenius law in Fig. 3c. The thermal gaps of the CIs are fitted to be 0.06 meV, 0.47 meV and 0.22 meV, for fillings of $-5$, $-6$ and $-7\ n_0$, respectively. This energy scale is relatively smaller than the theoretically calculated bandwidth as shown in Fig. 2b-c. It is noticed that the thermal gaps extracted (less than 1 meV) are also small as compared to the energy scale of the temperature range used for extracting them. As in our experiment, the thermal activation gaps are obtained from the temperature range of 10–50 K. This is mainly because those "gaps" are not as robust as band insulators, and are often defined as incipient gaps due to relatively weak-interaction strength. In transport experiment, they turn out

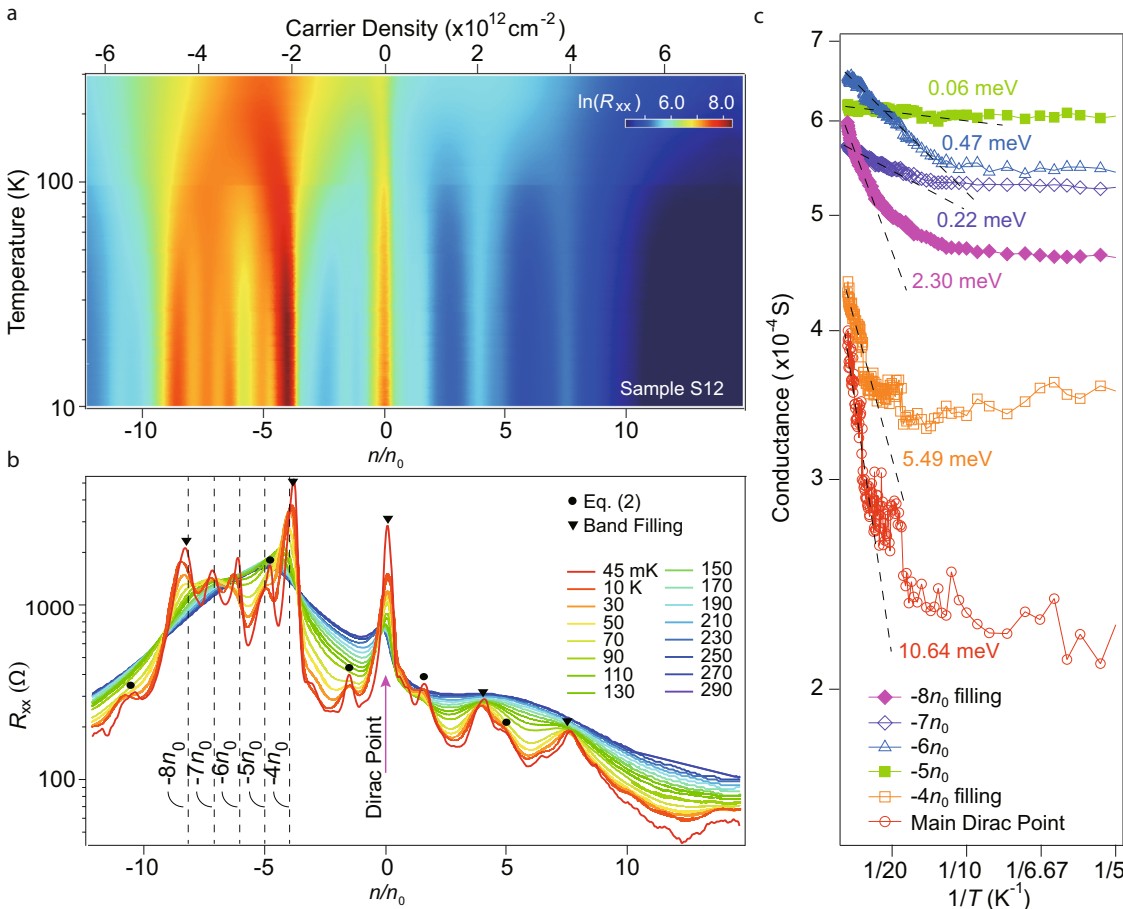

**Fig. 3 Temperature dependence of field effect in a doubly aligned h-BN/graphene/h-BN. a** Colour map of the sample resistance as a function of gate doping and temperature for sample S12. **b** Line cuts of data in (**a**) at different temperatures. Black solid triangles and black solid circles denote peaks identified from band filling and Eq. (2), respectively. **c** Arrhenius plot of representative resistive peaks in (**a**), with their fitted thermal excitation gaps indicated.

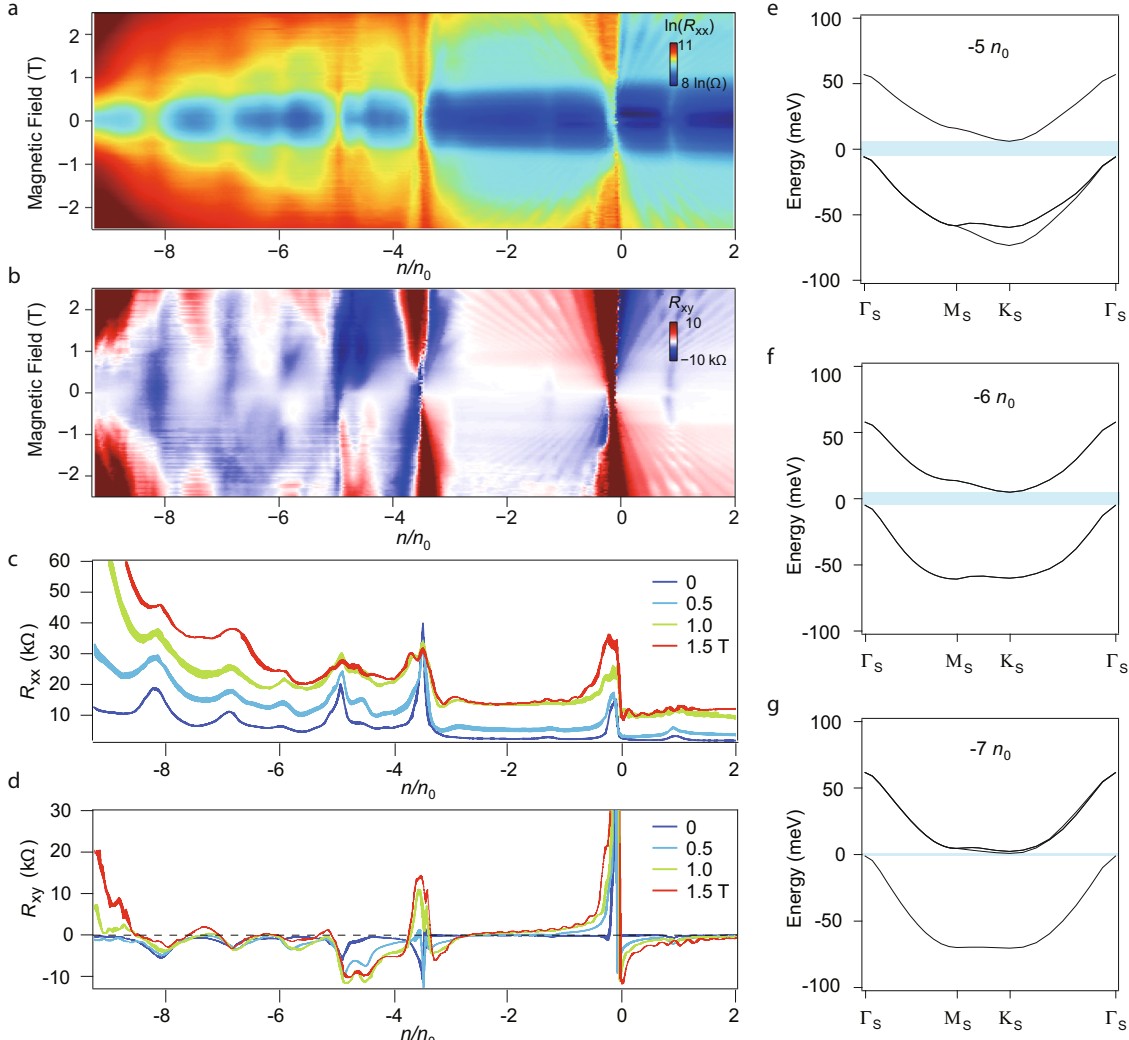

**Fig. 4 Correlated insulating states in h-BN/graphene/h-BN doubly aligned composite moiré. a, b** Colour maps of $R_{xx}$ (shown in a log scale for better contrast) and $R_{xy}$ measured in sample S60, scanned in the space of carrier density and magnetic field. **c, d** Line profiles of the data in (**a**) and (**b**) for several magnetic fields. Data obtained at $T = 50$ mK. **e–g** Band structure calculated using Hartree–Fock approximations for each integer fillings, which takes into account the inter-valley coherence orders. Gap openings (indicated by the light-blue regions in (**e**)–(**g**) of similar order of magnitudes as the width of the target band are seen at all integer fillings of $-5$, $-6$ and $-7$ $n_0$.

to be weak resistive peaks that cannot develop into more insulating ground state, and become saturating at temperatures below a few Kelvin. In other recently reported studies on twisted graphene systems, similar temperature ranges are often used for the fitting of thermal activation gaps for such weak resistive peaks[15,27]. Moreover, among four samples (S4, S12, S60 and S63) that exhibit similar resistive peaks at integer fillings from $-5$ to $-7$ $n_0$, the thermal activation gaps are extracted to be rather consistent, as shown in Supplementary Fig. 17, which speaks the reproducibility of the observed phenomenon.

We further performed, in a typical sample (S60), the zoomed-in scan of these CI states in the parameter space of carrier density and magnetic field, with the longitudinal ($R_{xx}$) and transverse resistance ($R_{xy}$) shown in Fig. 4a, b, respectively. Line cuts of $R_{xx}$ and $R_{xy}$ at different magnetic fields are plotted in Fig. 4c, d. It is noticed that in the band from $-4$ $n_0$ to $-8$ $n_0$, insulating states are seen at each integer fillings, with positive longitudinal magnetoresistances at finite magnetic fields. Direct comparisons of samples showing similar CI behaviours are shown in Supplementary Figs. 18, 19. We attribute the pseudogaps to strong Coulomb interaction, which lifts the degeneracy and induce the

correlating gaps, at fillings of $-5$, $-6$ and $-7$ $n_0$, as will be discussed in the coming text.

To address the origin of this correlating physical phenomenon, we performed theoretical calculations for those correlated gaps using Hartree–Fock methods (see "Methods"). In our calculations, one target band per valley per spin is taken into consideration, thus total four energy bands are involved in the Hartree–Fock analysis. Depending on whether valley $U(1)$ symmetry is broken or not, a gapped state driven by Coulomb interactions can be divided into two types: in the first type the gap is induced by valley and/or spin polarization, while in other type originates from an inter-valley coherent (IVC) order that couples the two valleys and breaks valley charge conservation (valley $U(1)$ symmetry) spontaneously. First, we consider the characteristic screened Coulomb interaction in this moiré system $U_M = e^2/(4\pi\varepsilon\varepsilon_0 L_M)$ (see more details in "Methods"), with $L_M$ being the moiré lattice constant in real space. It is noticed that, using the above relation, at a twist angle $\theta_G = 0.5°$, the amplitude of $U_M$ is only ~28 meV, which is roughly the energy scale of the valley and spin splittings if a valley- and/or spin-polarized state were the ground state at partial integer fillings. However, this value is still

far lower than the bandwidth of band II (~80 meV) shown in Fig. 2c. Thus, we conjecture that an IVC order is needed in order to open a global gap for such relatively weak Coulomb interaction strength (notice that other hypothesis of gap openings from such as Mott insulator on triangular lattice, or from strong Coulomb interaction-induced valley splitting, can be ruled out, as we are in a weak-interaction regime). As shown in Supplementary Fig. 20, it is seen that the IVC orders are existing at all fillings, with each average amplitude to be 1.42, 1.90 and 1.35 meV for fillings of $-5$, $-6$ and $-7$ $n_0$, respectively. However, it is worthwhile to note that at all of the three fillings, the IVC order amplitudes are highly dependent on the moiré wavevector, which can reach a maximal value of 50–90 meV at some small regions in the moiré Brillouin zone (see Supplementary Fig. 20), which is comparable to the bandwidth shown in Fig. 2c. Hence, thanks to these peaks of IVC orders, correlated states with notable energy gaps can occur at low temperatures in spite of the relatively weak Coulomb interaction strength and large bandwidth.

As shown in the calculated energy bands, when taking IVC order into account at $\theta_G = 0.5°$, in Fig. 4e–g, global gaps openings can be seen as indicated by the light-blue regions, with the value of 11.8, 9.7 and 1.8 meV, for $\nu = -5$, $-6$ and $-7$ $n_0$, respectively. The calculated correlated gaps are larger than the experimentally extracted thermal activation ones, which may come from the underestimated quantum fluctuations (thus overestimated gaps) in the Hartree–Forck simulations. It is known that the gap itself also strongly depends on experimental details, such as errors in the twist angle, and the exact dielectric constant in the materials. Nevertheless, the general trend of the three correlated gaps are in qualitative agreement with the ones obtained in Supplementary Fig. 17. We further obtain the global correlated gap sizes as a function of twist angle $\theta_G$, shown in Supplementary Fig. 21. It is seen that the global gap can reach up to the order of 10–20 meV for the perfectly aligned situation. However, considering the error of twist angle in the dual moiré sample fabrications, the global gap sizes can be much reduced. In the mean time, it is also found that the global gap sizes drastically diminish with an increased dielectric constant, as shown in Supplementary Fig. 22. On the other hand, to observe the correlated states at fillings of $-5$, $-6$ and $-7$ $n_0$ in the dual-moiré h-BN/graphene/h-BN devices, one has also to minimize as much as possible the inhomogeneities, as distortion of the superlattice can be seen in the AFM scans in Supplementary Fig. 11. Finally, we have to emphasize that the doubly aligned moiré superlattice in the studied system plays a crucial role to reduce the bandwidth into an appropriate scale (a few tens of meV of band II, as calculated in Fig. 2c, while band I is still too large with a width exceeding 100 meV, and band III may be too much influenced by the hybridization of lower energy bands). Overall, the concurrence of a doubly aligned moiré superlattice, and the hypothesis of the existence of IVC orders, together with the observed correlated states at fillings of $-5$, $-6$ and $-7$ $n_0$ in multiple dual-moiré samples, are self-consistent in this work.

We have demonstrated a series of monolayered graphene samples with a dual moiré constructed from both top and bottom h-BN/graphene interfaces. Thanks to the lattice symmetry tuned by the composite double moiré super potential, the system exhibits tunable band structure, with minibands at the order of a few tens meV for perfect alignment analysed using tight-binding and continuum models. By experimentally controlling the small twist angles of the doubly aligned superlattice, the signatures of the accompanied high order resistive peaks can indeed be sensitively tuned in multiple samples. In a best-aligned sample, the experimental observations agree with the picture of a hole-side filling sequence of each moiré minibands with $-4$, $-8$ and $-12$ $n_0$, in resemblance to the perfect align scenario. We also found correlated insulating states at integer fillings at $-5$ $n_0$ to $-7$ $n_0$ in typical doubly aligned single layer graphene samples S12, S4, S60 and S63. This correlated

behaviour is further attributed to a concurrence of the doubly aligned moiré superlattice and the existence of inter-valley coherent orders in the weak coupling regime. Our results suggest that composite super-moiré lattices with a dual twist from both graphene/h-BN interfaces can be a platform for the effective tuning of moiré minibands, and may provide insights for future correlated states engineering in such systems.

## Methods

**Device fabrication and measurements.** The BN-encapsulated graphene heterostructures were fabricated in ambient condition, using the dry-transfer method, with the crystals aligned to each of their exfoliated straight edges by a mechanical rotator. A Bruker Dimension Icon AFM was used for thicknesses and morphology tests. Raman measurements were performed by an HR 800 JobinYvon Horiba Raman spectroscopy. The electrical performances of the devices were measured using a BlueFors LD250 at mK temperature, a Cryogenics 4K system for temperature scannings, and a probe station (Cascade Microtech Inc. EPS150) for room temperature tests. Standard 4-probe measurements were applied with the low frequency lock-in technique.

**Band structure calculations.** For band structure calculations of the moiré superlattice, we employed a tight-binding (TB) model. Only $p_z$ orbitals were considered since only the states near the Fermi level are of interest. The TB Hamiltonian has the form[31]

$$H = \sum_i \epsilon_i |i\rangle \langle i| + \sum_{i \neq j} t_{ij} |i\rangle \langle j|, \tag{3}$$

where $|i\rangle$ is the $p_z$ orbital of the atom located at $\vec{r}_i$, $\epsilon_i$ is the onsite energy, and $t_{ij}$ is the coupling parameter. For the h-BN/graphene/h-BN heterostructure, both $pp\pi$ and $pp\sigma$ interactions are important. According to the Slater-Koster formula[32], $t_{ij}$ is given by:

$$t_{ij} = n^2 V_{pp\sigma}(r_{ij}) + (1 - n^2) V_{pp\pi}(r_{ij}), \tag{4}$$

where $n$ is the direction cosine along the $z$-direction of the vector $\mathbf{r}_j - \mathbf{r}_i = \mathbf{r}_{ij}$, and $r_{ij} = |\mathbf{r}_{ij}|$. Following previous studies[33,34], the functions $V_{pp\pi}$ and $V_{pp\sigma}$ are assumed to have the following forms:

$$V_{pp\pi}(r_{ij}) = V_{pp\pi}^0 e^{q(1 - r_{ij}/d_0)} \quad V_{pp\sigma}(r_{ij}) = V_{pp\sigma}^0 e^{q(1 - r_{ij}/a_0)}. \tag{5}$$

Here, $V_{pp\pi}^0 = 0.48$ eV and $V_{pp\sigma}^0 = -2.7$ eV are the hopping integrals for the $pp\pi$ and $pp\sigma$ interactions, respectively. $q = 2.218$ is chosen so that the next-nearest intralayer hopping is $0.1 V_{pp\sigma}^0$. $a_0 = 1.418$ Å is the C-C bond length in graphene, and $d_0 = 3.35$ Å is the spacing between the graphene and h-BN layers. The intra- and inter-layer hoppings are truncated with a 6 Å cutoff. The onsite energies for C, B and N are 0 eV, 3.34 eV and $-1.40$ eV, respectively. These parameters have been widely used to study bilayer graphene and graphene/BN systems[23,29,33].

We consider the inter-site Coulomb interaction $H_C$ as:

$$\frac{1}{2N_S} \sum_{\alpha\alpha'} \sum_{\mathbf{k}\mathbf{k}'\mathbf{q}} \sum_{\sigma\sigma'} V(\mathbf{q}) \hat{c}_{\mathbf{k}+\mathbf{q},\alpha\sigma}^+ \hat{c}_{\mathbf{k}'-\mathbf{q},\alpha'\sigma'}^+ \hat{c}_{\mathbf{k}',\alpha'\sigma'} \hat{c}_{\mathbf{k},\alpha\sigma} \tag{6}$$

for Hartree–Fock mean-field calculations in the moiré system. Here $\alpha$ refers to layer and sublattice indices. $\mathbf{k}$ and $\mathbf{q}$ are atomic wavevectors expanded around the Dirac points, and $\sigma, \sigma'$ are the spin indices. We choose the double-gate screened Coulomb interaction, which is expressed as $V(\mathbf{q}) = e^2 \tanh(\mathbf{q}d_s)/(2\Omega_M \varepsilon \varepsilon_0 \mathbf{q})$. Here $\varepsilon \sim 4.0$ is background dielectric constant, and will be further changed as a variable in our calculations. $d_s = 200$ Å is the screening length. $\Omega_M$ is the area of moiré Brillouin zone. The screened Coulomb interaction can be re-written as $V(\mathbf{q}) = U_M q_M \tanh(\mathbf{q}d_s)/\mathbf{q}$ in which $q_M = 4\pi/\sqrt{3}L_M$ is the length of reciprocal lattice vector, and $U_M = e^2/(4\pi\varepsilon\varepsilon_0 L_M)$ is characteristic Coulomb interaction. It is noted that at the twist angle of $\theta_G = 0.5°$, the amplitude of $U_M$ is only ~28 meV which is far lower than the bandwidth of band II (~80 meV).

In this system, the Coulomb interactions can be divided into the intra-valley term $H_C^{intra}$ and the inter-valley term $H_C^{inter}$,

$$H_C^{intra} = \frac{1}{2N_s} \sum_{\alpha\alpha',\mu\mu',\sigma\sigma'} \sum_{\mathbf{k}\mathbf{k}'\mathbf{q}} V(\mathbf{q}) \\ \times \hat{c}_{\mathbf{k}+\mathbf{q},\mu\sigma\alpha}^\dagger \hat{c}_{\mathbf{k}'-\mathbf{q},\mu'\sigma'\alpha'}^\dagger \hat{c}_{\mathbf{k}',\mu'\sigma'\alpha'} \hat{c}_{\mathbf{k},\mu\sigma\alpha} \tag{7}$$

$$H_C^{inter} = \frac{1}{2N_s} \sum_{\alpha\alpha',\sigma\sigma'} \sum_{\mu} \sum_{\mathbf{k}\mathbf{k}'\mathbf{q}} V(|\mathbf{K} - \mathbf{K}'|) \\ \times \hat{c}_{\mathbf{k}+\mathbf{q},\mu\sigma\alpha}^\dagger \hat{c}_{\mathbf{k}'-\mathbf{q},-\mu\sigma'\alpha'}^\dagger \hat{c}_{\mathbf{k}',\mu\sigma'\alpha'} \hat{c}_{\mathbf{k},-\mu\sigma\alpha} \tag{8}$$

Here $H_C^{intra}$ represents the process of two electrons created and annihilated in the same valley, and $H_C^{inter}$ means that two electrons are created in $\mu$ and $-\mu$ and get annihilated in $-\mu$ and $\mu$ valleys with the interaction energy $V(|\mathbf{K} - \mathbf{K}'|)$. We only consider the intra-valley interaction because the inter-valley interaction is two orders of magnitudes weaker than intra-valley interaction. With the Coulomb

interaction projected onto the band II, we start with 32 possible initial trial wavefunctions in the valley-spin-sublattice space, for example, spin-polarized/valley-polarized/IVC cases, and perform the self-consistent calculations to solve the interacting Hamiltonian with Hartree–Fock approximations. Finally, we find the solution with the lowest total energy as the Hartree–Fock ground state.

## Data availability

The data that support the findings of this study are available at Zenodo, https://doi.org/10.5281/zenodo.5078093.

## Code availability

The computational codes that support the findings of this study are available from the corresponding authors upon reasonable request.

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

## Acknowledgements

This work is supported by the National Key R&D Program of China (2017YFA0206302, 2019YFA0307800, 2020YFA0309601, and 2018YFA0306900), and is supported by the National Natural Science Foundation of China (NSFC) with Grants 11974357, U1932151, 51627801, 12074029, U1930402, 52031014, and 11991060. S.Z. and J.L. acknowledge the start-up grant of ShanghaiTech University. X.L. acknowledges support from the Joint Research Fund of Liaoning-Shenyang National Laboratory for Materials Science with Grant No. 2019JH3/30100031. T.Y. acknowledges support from Liaoning Provincial Natural Science Fund with Grant 2021-MS-006. K.W. and T.T. acknowledge support from the Elemental Strategy Initiative conducted by the MEXT, Japan (Grant Number JPMXP0112101001), JSPS KAKENHI (Grant Numbers 19H05790 and JP20H00354) and A3 Foresight by JSPS.

## Author contributions

Z.H. conceived the experiment, and Z.H., Z.Z., J.L., Y.Y. and J.K. supervised the overall project. Z.L., J.K. and T.Y. carried out continuum and tight-binding modellings and simulations. X.S., H.Z., J.H., X.L., and Z.W. fabricated the devices. J.L. and S.Z. carried out band structure calculations of the correlated states. X.S., Z.H., K.Y. and Y.Y. carried out electrical transport measurements; K.W. and T.T. provided high quality h-BN bulk crystals. X.S. and Z.H. analysed the data, with J.L., J.K., T.Y., Y.Y., S.Z., J.M. and M.Z. participated in thorough discussions. The manuscript was written by Z.H., X.S., J.L., Z.L. and J.K., with discussion and inputs from all authors.

## Competing interests

The authors declare no competing interests.
