## [Peer Review File · Nature Communications]

REVIEWER COMMENTS

Reviewer #1 (Remarks to the Author):

The authors investigate a hBN/graphene/hBN structure where the hBN layers introduce moire patterns in graphene. Such two moire patterns can create a new moire pattern that is called supermoire. The latter has a much larger unit cell and consequently it should be possible to enhance correlation effects (because lower densities are needed).

Pseudogaps are observed and claims are made about possible correlated insulating states.

The paper is timely and of interest.

I have found a number of issues that should be considered before one can decide about the publication of this manuscript.

1) The manuscript should be improved by a careful reading. There are several corrections to be made. For example:

- Just below Fig. 2, left column: 'filling fraction ν ' is not defined. Or should it be n/n_0 ?

- Below Fig. 4 left column: Fig. 3e should be Fig. 2e; 'integer fillings at integer fillings'

2) The thermal gaps obtained from Fig. 3b differ only with at most 20% from the average. This is surprising because of the very different origin of some of the gaps. Can the authors give an explanation for this? Why do the authors determine the energy gap from the high temperature range? The temperature behaviour of the different peaks can be very different. What is the reason for that? For example 5, 6, 7 fillings are claimed to be possible correlated insulating states. However the temperature dependence of 5,6 is very different from 7. I am very sceptical about the determination of the energy gaps from Fig. 3b. In fig. 3a(bottom) it appears that all the separate peaks are gone for $T > 130\text{K}$.

3) In Figs. 4(c,d,e): why does R_{xy} not change sign when $B \rightarrow -B$?

4) The authors should stress what is the difference between the present paper and Ref. 16 where almost the same system was investigated. It appears to me that the physics is very similar.

5) The supplementary information should be carefully checked.

- Something is missing in the first sentence of the last paragraph on p. 2.

- p. 5: last sentence makes reference to Fig. 2(g-h) which does not exist.

- Fig. 2a: it appears that there is a small vertical shift in the I-V curve at $V_{ds}=0$. Is this related to superconductivity?

- Fig. 4: which temperature range is considered?

- Some figure captions are missing some important information. E.g. Figs. 8 and 9: results are for which temperature?

Reviewer #2 (Remarks to the Author):

The authors study magneto transport of hBN/graphene/hBN consisting of two sets of Moire pattern, which the author refers to as "dual moire". The exact system has been previously studied extensively, both transport-wise and structural-wise.

To name a few:

[1] Finney, N. R. et al. Nature Nanotechnology volume 14, pages1029–1034 (2019).

[2] Nano. Lett. 19, 2371–2376 (2019).

[3] Sci. Adv. 5, eaay8897 (2019).

In these previous works (which the author cited), the studies were more comprehensive with capability to dynamically tune the twist angle with AFM tips, and direct measurement of dual Moire patterns. The prior works refers to the complicated magneto transport data as a consequence of "a highly reconstructed graphene band structure featuring multiple secondary Dirac points" similar to Hofstadter butterfly in absence of correlation. The completeness and scope of this manuscript, in comparison, is limited and provides little new information to the exact same research topic. The

claim of "correlated states" is also not well-justified with the data presented, compared to prior works (particularly reference 1) where similar data were shown.

I think the novelty, potential impact, and soundness of the claim made does not meet the high bar of Nature communication, and additional data and significant revision of the narrative presented are necessary to strengthen the paper. Specifically, I recommend the authors to consider the following suggestions:

1. The lack of direct characterization of associated wavelength, making the whole interpretation based on somewhat shaky grounds. It would have been however feasible to measure the moiré wavelength (and the beating pattern of the two Moire patterns) by performing AFM after each pick-up step in the fabrication process, as was done by Wang et al. in *Sci. Adv.* 5, eaay8897 (2019). Such a characterization would have made the interpretation of the results much more robust. (*Nano. Lett.* 19, 2371–2376 (2019) and *Sci. Adv.* 5, eaay8897(2019); these references should appear in the manuscript). The "filling factor" the author is labeling in their main results, shows quite weak temperature dependence only at -5, -6, -7 ν_0 , and missing features at other integer fillings (-1, -2, -3 etc). It is therefore useful to directly verify the Dual moire lengthscale aside from transport.

2. Similar to Hofstadter butterfly, when two lattice with small twist angle form a beating pattern (or Moire pattern), avoided-crossing happens, gap opens and resistive peaks arises at the carrier density corresponding to the periodicity of the superlattice. The author fig. 1d showed just that, from one set of Moire at hBN/graphene interface. This is just basic solid-state physics at play and the gap is a consequence of avoided-crossing of single-electron bands instead of correlation. Zooming-into smaller range (fig.3) reveals peaks that could correspond to higher order Dual Moire, but the insulating behavior as well as the extracted gap can simply be a result of avoided-crossing as well in absence of correlation, which has been reported multiple times in similar previous studies.

In addition to verification of filling factors (see my comment 1), it would strengthen the claim of "correlated" states if the author can demonstrate co-existence of a superconducting dome next to the correlated insulating states (or at least a metallic state developing into a superconducting dome), for example, similar to that of reference [3] in the manuscript "Cao, Y. et al. Unconventional superconductivity in magic-angle graphene superlattices. *Nature* 556, 43–50 (2018)." The feature between -5 and -6 ν_0 is still far from zero resistance and the temperature dependence of it can be a consequence of peaks at -5 and -6 ν_0 .

3. Lastly, at least data from one more control sample (in addition to sample-4 and 12) should be presented in the manuscript and demonstrate reproducibility, with a full set of magneto transport similar to data from S12 data and plotted side-by-side for direct comparison (figure 3 and 4).

4. Minor detail: Reference [28] and [31] are the same paper that appeared twice as separate references.

Reviewer #3 (Remarks to the Author):

The authors investigate transport properties of monolayer graphene aligned to both top and bottom encapsulating hexagonal boron nitride (h- BN). They report multiple conductivity minima away from the main Dirac point, which are interpreted by the emergence of higher energy band. In addition, the authors report evidence of correlated insulating states at integer fillings of -5, -6, -7 electrons per moiré unit cell.

This paper is highly relevant to the topic of moire physics, which has generated intense interests in the field of condensed matter and material engineering research. It is recently demonstrated that aligning graphene with hBN generates unexpected physical structures. For example, ferromagnetism emerges from the moire lattice when ABC trilayer graphene or magic-angle tBLG

are aligned with hBN. Even bernal bilayer graphene, which has been studied extensively in prior efforts, exhibits unexpected ferroelectricity when aligned with hBN. Based on these recent developments, one could argue that it is of great interest to examine the influence of doubly aligned hBN/graphene/hBN alignment.

Indeed, the authors report a few interesting results in their transport measurements. The multiple resistance peaks away from the main Dirac point offers evidence that multiple energy bands are accessible by electrostatic doping. If this is due to the band flattening effect of the double alignment between graphene and hBN, it could feature strong electron correlation, which will attract future efforts studying this structure. In addition, incipient insulating states are shown to emerge at almost every integer filling of the moire unit cell. These states are interpreted as correlated insulators stabilized by Coulomb interaction, consistent with flat energy band and strong correlation. Overall, I find the manuscript to be innovative and well written. It deserves publication in Nature communication, if the authors could address the following comments.

(a) when graphene is aligned with both top and bottom hBN, two moire patterns emerge, between graphene and top (bottom) hBN. As the twist angle is usually slightly different, these two moire lattice feature different lattice constant, giving rise to two sets of satellite peaks, one from each moire, as shown in Ref. [15]. Is the same phenomenon observed here?

(b) related to the last question, how are the twist angles determined in Fig.2e? For example, sample 12 is labeled with 0 and 0.5 degrees. I assume these angles are determined based on the position of respective satellite peaks, and it will be helpful to show these discussions in the main text.

(c) while CIs are observed at 0/0.5 degree, no CIs at 0.45 degrees. How big is the error bar in determining twist angles?

(c) in Fig.3 and 4, a series of CIs are shown to appear at integer fillings. CIs between filling -4 to -8 are particularly robust compared to other filling. Does this reflect properties that are specific to band II? If so, it should be discussed in the manuscript.

(d) Fig.4 tried to argue for an insulating feature along the red dashed line, while the width of the feature is bigger than the separation between the red and white dashed lines. Why is this data set only measured up to 2.5T? A Landau fan measurement at higher field could be more convincing for the point the authors are trying to make here.

A few more minor points:

(1) the energy gap labelled in Fig.3b appears to be wrong. For example, the slope of the 9.05meV line is smaller than that of the 6.82meV.

(2) in the second paragraph of the RESULTS section, the perfect-alignment regime is used without definition (it is defined several sentences later. It will be helpful to define the term sooner). The term perfect alignment is also misleading. It sounds like perfect alignment (0 degree), even though it refers to <1degree. Maybe "alignment regime" is more appropriate.

(3) left column on page 5, typo in "due to possible existing stains..."

Response to Reviewer #1:

General Comment: *The authors investigate a h-BN/graphene/h-BN structure where the h-BN layers introduce moiré patterns in graphene. Such two moiré patterns can create a new moiré pattern that is called super moiré. The latter has a much larger unit cell and consequently it should be possible to enhance correlation effects (because lower densities are needed).*

Pseudo-gaps are observed and claims are made about possible correlated insulating states.

The paper is timely and of interest.

Response: We thank Reviewer #1 for her/his positive comments. We present here answers to her/his questions in the following, and also have modified the manuscript accordingly. Modified parts are highlighted in blue color in the SI or main text in the revised version of our manuscript.

References/figures appearing in this Response-to-Referees file will be re-indexed if any of them are added in the revised main text (and the Suppl. Info.) of our manuscript.

Comment 1: *“I have found a number of issues that should be considered before one can decide about the publication of this manuscript.*

1) The manuscript should be improved by a careful reading. There are several corrections to be made. For example:

- Just below Fig. 2, left column: 'filling fraction ν ' is not defined. Or should it be n/n_0 ?

- Below Fig. 4 left column: Fig. 3e should be Fig. 2e; 'integer fillings at integer fillings'”

Response: We appreciate the very careful reading of our manuscript. Filling fraction ν was indeed not well defined in the left column below Fig.2 in the main text. These are actually Landau index for each satellite peaks, i.e., at each filling of 0, -4, -8, and -12 n_0 , there is a set of Landau fan, with their own filling fractions to be $\pm 2, \pm 6, \pm 10$, etc. The ‘filling fraction’ ν corresponds to the ν in Eq. (1) in the main text:

$$\frac{n}{n_0} = \nu \frac{\phi}{\phi_0} + s$$

Following the reviewer’s suggestions, we have now updated the term “filling fractions” in the left column below Fig.2 with “topological index ν ”, to avoid

confusion.

The typo of “Fig. 3e” in the left column below Fig. 4, has been changed into “Fig. 2e”. In the mean time, the repeated words “integer fillings at integer fillings” has been corrected also. The corrected phrases are highlighted in blue color in the revised manuscript.

We appreciate very much for these details pointed out by our referee.

Comment 2: *“The thermal gaps obtained from Fig. 3b differ only with at most 20% from the average. This is surprising because of the very different origin of some of the gaps. Can the authors give an explanation for this? Why do the authors determine the energy gap from the high temperature range? The temperature behaviour of the different peaks can be very different. What is the reason for that? For example 5, 6, 7 fillings are claimed to be possible correlated insulating states. However the temperature dependence of 5,6 is very different from 7. I am very skeptical about the determination of the energy gaps from Fig. 3b. In Fig. 3a (bottom), it appears that all the separate peaks are gone for $T > 130\text{K}$.”*

Response: The reviewer raised a great point here, and we really appreciate her/his comments. In fact, in many of the systems with thermally activated gaps, the resistance versus temperature (let’s call it R - T curve) can be very different in different temperature regimes. It is therefore rather critical in choosing the temperature range while extracting the thermal gaps. In the twisted graphene systems, there are a few reports that study the correlated insulating gaps.

However, in our doubly-aligned h-BN/graphene/h-BN system, the temperature dependence of the pseudo-gaps at fillings of -5, -6, and -7 n_0 are developed only at very low temperature regime (i.e., below 50 K as shown in Fig. 3a in the main text). Even in this case, the temperature dependences are quite weak. We thus fully agree with the referee that it is very important to re-perform the thermal gap extractions in the lower temperature range rather than the previously used one (50 to 250 K, in the 1st submission), where the results seem to be more coherent.

By fitting the thermal activation gaps in the temperature range of 5 to 50 K, as shown in FIG. R1 below, one obtains thermal gaps of about 10.64 meV for the main Dirac point, while those for fillings of -4 and -8 n_0 are estimated to be 5.49 and 2.30 meV, respectively. Notably, for the correlated insulating gaps at fillings of -5, -6, and -7, the thermal gap are estimated to even smaller, with each values to be 0.06, 0.47, and 0.22 meV, respectively.

The updated figure is now included in the revised manuscript in Fig. 3b, with the descriptions updated in the main text, as well.

FIG. R1. Thermal activation gaps at integer fillings. Conductance of the sample at different fillings plotted in a log scale, as a function of $1/T$. Here, the temperature range is set to be 5 to 50 K.

In the non-interacting picture, the doped system is metallic at the fillings of $\nu = -5$, -6 and $-7 n_0$. So we attribute the pseudo-gaps to strong Coulomb interaction at these fillings, because the Coulomb interaction can lift the degeneracy and induce the correlating gaps. To explain this correlating physical phenomenon, we performed theoretical calculations for those correlated gaps (-5 , -6 , $-7 n_0$) using Hartree-Fock methods.

There are always two types of gaps induced by the Coulomb interaction. The first type is induced by valley or spin polarization, and the other type originates from inter-valley coherent interaction. Our calculations reveal that the strength of polarization (valley polarization or spin polarization) is smaller than target band's width, so inter-valley coherent interaction is necessary to open the global gap.

As shown in FIG. R2, it is seen that the inter-valley coherent (IVC) orders are calculated to be existing at all fillings, with the amplitude of each to be 1.42, 1.90, and 1.35 meV for fillings of -5 , -6 and $-7 n_0$. Thanks to these IVC order parameters, insulating behavior can occur at low temperatures. It thus explains why the CIs are

still taking place even though the interaction-induced valley or spin polarization is not strong enough to open global gaps. In the mean time, the doubly aligned moiré superlattice also helps to reduce the band width into an appropriate scale (a few tens of meV of band II, as calculated in Fig. 2c in the main text). Overall, the concurrence of a doubly aligned moiré superlattice, and the existence of IVC orders, is the origin of the observed correlated states at fillings of -5 , -6 , and $-7 n_0$, in our work.

We notice that in Comment 2 from Referee#2, related question about the observed correlated pseudo-gap is also raised there, and we will discuss it again in that section. Discussions about the IVC calculations described here have been included in the revised version of Supplementary Information.

FIG. R2. Calculated correlated gaps occurring at fillings of -5 , -6 and $-7 n_0$. (a)-(c) illustrate the calculated gap openings from inter-valley coherence, with the amplitude of each IVC order to be 1.42, 1.90, and 1.35 meV for fillings of -5 , -6 and $-7 n_0$.

Comment 3: “In Figs. 4(c,d,e): why does R_{xy} not change sign when $B \rightarrow -B$?”

Response: We thank Reviewer#1 for her/his constructive suggestion. It is indeed usually expected to see a sign change in the R_{xy} when magnetic field is swapped in its vertical direction (i.e., which is further changed in the sign when doping is shifted from holes to electrons, or vice versa).

In the revised manuscript, we have fabricated new doubly-aligned

h-BN/graphene/h-BN samples. First, we screen samples by measuring the field effect as a function of temperature, where the major pseudo-gaps should clearly develop at ground states (the sample yield is about 1/20) if the twist angles of both top and bottom h-BN with respect to the middle graphene layer is small enough (a simple criteria would be that the left satellite peak is over 1.3 times more resistive than the main Dirac peak).

FIG. R3. Temperature dependence of field effect curves (R_{xy}) for sample S-56. Similar behavior as S-12 and S-4 is observed, with pseudo-gaps developing at fillings of -5 , -6 , and $-7 n_0$ at temperatures below 100 K.

FIG. R4. R_{xy} of Sample S-56 at different magnetic fields. Clear sign changes in the main Dirac point, as well as at fillings of -4 and $-6 n_0$ can be seen, where there is a change of carrier type from electrons to holes. Data measured at 50 mK.

Among the new devices, Sample S-56 gives similar behavior as that of S-12 and S-4, as shown in FIG. R3. Moreover, R_{xy} of sample S-56 shows clear sign changes in the main Dirac point, as well as at some satellite peaks where there is a change of carrier type from electrons to holes, shown in FIG. R4. We have updated the Fig.4 with new R_{xy} data in the revised manuscript.

We also notice that the response in this section is overlapping with the Comment 4 from the Referee#2, therefore we will discuss in that section again on the data obtained in newly fabricated devices in this 2nd submission.

Comment 4: “The authors should stress what is the difference between the present paper and Ref. 16 where almost the same system was investigated. It appears to me that the physics is very similar.”

Response: We thank the referee for her/his suggestion. Indeed, Ref. [16] (i.e., Nano. Lett. **19**, 2371-2376 (2019)) deals with the system where there are twist angles (θ_a and θ_b) from both top and bottom h-BN, with respect to the encapsulated middle graphene layer. As discuss in Ref. [16], also their topic figure is cited and shown below as FIG. R5, three sets of moiré super lattice will be formed in this generic situation. ϕ_1 and ϕ_2 are used in their work to denote top and bottom twist angles.

FIG. R5. Illustration of three different moiré super lattices formed in a h-BN/graphene/h-BN heterostructure.^[R1] Blue, red colors denote top and bottom h-BN, while black color denotes the encapsulated middle graphene layer. Figure adapted from Ref. [R1] (Ref. [16] in the main text).

The difference between the present study and that in Ref. [16] is that we consider

^[R1] Wang, Lujun, *et al.* Nano Lett. **19**, 2371-2376 (2019).

the cases where ϕ_1 and ϕ_2 are very close to zero, where flat bands (with width of a few tens of meV) appear in the aligned regime, as shown in the simulated results in Fig. 2b-c in the main text. Only in the aligned regime, the correlated gaps at fillings of -5, -6, and -7 n_0 can then be observed. Furthermore, as we described in the response to the Comment 2 above, we performed new theoretical calculations that suggest the origin of correlated states to be the inter-valley coherence enhanced electron interaction.

As a summarization, the geometric model (see FIG. R5) studied in Ref. [16] is the same as ours, but the physics of correlated states at the very small twisting angle (in the vicinity of 0-0-0 degree alignment) is absent in their work.

Comment 5: *“The Supplementary Information should be carefully checked.*

- *Something is missing in the first sentence of the last paragraph on p. 2.*
- *p. 5: last sentence makes reference to Fig. 2(g-h) which does not exist.*
- *Fig. 2a: it appears that there is a small vertical shift in the I-V curve at $V_{ds}=0$. Is this related to superconductivity?*
- *Fig. 4: which temperature range is considered?*
- *Some figure captions are missing some important information. E.g. Figs. 8 and 9: results are for which temperature?”*

Response: We really appreciate the referee for her/his careful readings of our Supplementary Information. 1) The missing phrase in the first sentence of the last paragraph on Page.2 has been corrected. 2) Sorry for our mistake, the last sentence in Page.5 is making its reference to Fig. 2(e). 3) The Supplementary Fig.2a was measured at room temperature, which should not be related to superconductivity. 4) Temperature range in Supplementary Figure 4 is now corrected. 5) We have added the temperature information in Supplementary Figure 8-9.

Response to Reviewer #2:

General Comment: *The authors study magneto transport of h-BN/graphene/h-BN consisting of two sets of Moiré pattern, which the author refers to as "dual moiré". The exact system has been previously studied extensively, both transport-wise and structural-wise. To name a few:*

[1] Finney, N. R. et al. Nature Nanotechnology volume 14, pages1029–1034 (2019).

[2] Nano. Lett. 19, 2371–2376 (2019).

[3] Sci. Adv. 5, eaay8897 (2019).

In these previous works (which the author cited), the studies were more comprehensive with capability to dynamically tune the twist angle with AFM tips, and direct measurement of dual Moiré patterns. The prior works refers to the complicated magneto transport data as a consequence of "a highly reconstructed graphene band structure featuring multiple secondary Dirac points" similar to Hofstadter butterfly in absence of correlation. The completeness and scope of this manuscript, in comparison, is limited and provides little new information to the exact same research topic. The claim of "correlated states" is also not well-justified with the data presented, compared to prior works (particularly reference 1) where similar data were shown.

I think the novelty, potential impact, and soundness of the claim made does not meet the high bar of Nature communication, and additional data and significant revision of the narrative presented are necessary to strengthen the paper. Specifically, I recommend the authors to consider the following suggestions.

Response: We are grateful for the very constructive comments from the referee#2. As also raised in the Comment 4 from Referee#1, the geometric model for doubly twisted moiré super lattice in h-BN/graphene/h-BN was indeed reported in the above 3 references (we also cited in our main text as Ref. [15-17]), which we will discuss a bit more in the following responses to the comments from the Referee#2.

For example, Finney, N. R. *et al.* Nature Nanotechnology 14, 1029–1034 (2019) (i.e., Ref.[15] in our manuscript) reported manipulation of twist-angle of h-BN on graphene, using an AFM tip. In this specific study, the authors assumed a 0-degree aligned graphene/h-BN bilayer substrate (i.e., $\theta_b=0$), and only the top twist angle θ_t is modulated, as indicated in FIG. R6.

However, that paper investigated transport measurements mainly for a device with 0-0-60 degree alignment, where the correlated physics is absent in that regime, since the flat bands are only expected at 0-0-0 alignment as indicated by our theoretical considerations. Nevertheless, it is very important that this study gave a very nice reference of field effect curves of all θ_t (while θ_b is set to be 0). In the scenario of perfect alignment ($\theta_t=\theta_b=0$), left satellite resistive peak is about 1.4 times

higher than the central Dirac resistive peak at 300 K, which is used as a practical criteria to select devices for cooling down in our work.

On the other hand, Nano. Lett. 19, 2371–2376 (2019), and Sci. Adv. 5, eaay8897 (2019) (which are Ref. [16-17] in our previous submission) mainly worked on geometric model, which can solve each satellite peak at a specific doping (originated from electron filling in the unit area of the moiré super lattices).

In short, the geometric model (see FIG. R5 in our response to the Referee#1 as well) studied in these reported results is the same as ours, but the physics of correlated states at the very small twisting angle (in the vicinity of 0-0-0 degree alignment) is absent in their works.

As to the “correlated states”, the Referee#2 mentioned that there should be more justifications to strengthen up this point. We fully agree with her/his opinion, and have performed both additional experimental and theoretical studies in this revision. As a summarization (and will be discussed in more detail in each of the following responses), we have fabricated new devices that show similar behaviors, as well as Hartree-Fock calculations to address the origin of the correlated electronic states, which is a consequence of inter-valley coherence (IVC) enhanced gap opening at fillings of -5, -6, and -7 n_0 .

We sincerely wish that our new data along with the updated discussion included in this revision will be of satisfaction to justify the novelty and difference of our work, as compared to those previously reported.

FIG. R6. Illustration of the doubly aligned h-BN/graphene/h-BN experiment reported elsewhere^[R2] (Ref [15] in the main text). In the study, the authors assumed a 0-degree aligned graphene/h-BN bilayer bottom substrate (i.e., $\theta_b=0$), and only the top twist angle θ_t is rotated using an AFM tip. Their low temperature transport measurements mainly focused on a device with 0-0-60 degree alignment, where the correlated physics is absent in that regime.

^[R2] Finney, N. R. *et al.* Nature Nanotechnology, **14**, 1029-1034 (2019).

We now will answer all the comments from Referee#2 in a point-to-point manner in the following paragraphs.

Comment 1: *“The lack of direct characterization of associated wavelength, making the whole interpretation based on somewhat shaky grounds. It would have been however feasible to measure the moiré wavelength (and the beating pattern of the two Moiré patterns) by performing AFM after each pick-up step in the fabrication process, as was done by Wang et al. in Sci. Adv. 5, eaay8897 (2019). Such a characterization would have made the interpretation of the results much more robust. (Nano. Lett. 19, 2371–2376 (2019) and Sci. Adv. 5, eaay8897(2019); these references should appear in the manuscript). The “filling factor” the author is labeling in their main results, shows quite weak temperature dependence only at -5, -6, -7 n0, and missing features at other integer fillings (-1,-2,-3 etc). It is therefore useful to directly verify the Dual moiré length-scale aside from transport.”*

Response: First, as the referee also mentioned in her/his general comments (the previous paragraph), the 3 relative works (i.e., *Nature Nanotechnology*, 14, 1029 (2019), *Nano. Lett.* 19, 2371 (2019) and *Sci. Adv.* 5, eaay8897(2019)) are already cited in the very first submission of our manuscript as Refs. [15-17]. Here, we appreciate the referee to verify it in this comment.

Second, we thank the referee very much for her/his very constructive suggestions. Indeed, a direct atomic force microscopy (AFM) scan will make the interpretation of our results much more robust. According to her/his suggestions, we have added characterizations of the doubly-aligned h-BN/graphene/h-BN heterostructure by performing AFM after each pick-up step in the fabrication process.

As shown in FIG. R7, we performed AFM characterizations (using a friction force mode, with an OXFORD qp-BioAC-20 tip. Similar method was reported elsewhere.^[R3]) directly on the first aligned graphene/h-BN on PPC substrate (the stack was flipped over with the sequence from top to bottom as: graphene/h-BN/PPC/PDMS/Glass-slide). It is seen in FIG. R7a&c that a moiré wavelength of $\lambda \sim 13.6$ nm was obtained, which is very close to a perfect alignment, where λ should be ~ 14 nm.^[R4] Then, the whole stack is used to continue for another pick up of h-BN. Here, we purposely chose very thin second h-BN flakes, since thick h-BN does not allow the measurement of buried interfacial moiré superlattice.

^[R3] Marsden, Alexander J. *et al.* *Nanotechnology* **24**, 255704 (2013).

^[R4] Yankowitz, Matthew, *et al.* *Nature physics* **8**, 382-386 (2012).

FIG. R7. Direct verification of the dual moiré length-scale. Moiré wavelength of about $\lambda \sim 13.6$ nm was seen after each pick-up step in the sample fabrication process. Scale bars in optical images in (a)-(b) are 10 μm .

FIG. R8. AFM scan of dual moiré length-scale obtained in another Sample. $\lambda \sim 16.6$ nm (with strains existing) was found in the first pick up, while $\lambda \sim 14.5$ nm was seen for the second pick up. Inhomogeneous regions can be seen in the red dashed box, indicating that the doubly aligned area can be rather local, and this is why the device yield is very low (less than 1/20). Scale bars in optical images in (a)-(b) are 10 μm .

As shown in FIG. R7b&d, a typical doubly aligned h-BN/graphene/h-BN heterostructure gives the second moiré wavelength of about 13.7 nm, very close to that obtained in the first picking up. This is in agreement with theoretic predicted value and can be a strong complementary to the transport measurements in the manuscript.

It is noticed that, some of the aligned h-BN/graphene samples are of slightly larger moiré wavelength than the theoretical maximum size. As shown in the AFM scan in FIG. R8a&c, in another sample, the moiré wavelength is found to be 16.6 nm, indicating an existing strain, which is in agreement with other reports.^[R5] After another picking up process, the resulted doubly-aligned h-BN/graphene/h-BN yields another moiré wavelength of 14.5 nm, shown in FIG. R8b&d. Moreover, clear inhomogeneities in the moiré superlattices can be seen in the red dashed box in FIG. R8d, indicating that the well aligned area can sometimes be rather local, with micronmeter sizes or smaller. This kind of inhomogeneities in moiré superlattice is also reported elsewhere.^[R6] This also speaks why the device yield is very low (less than 1/20) in our study.

The above AFM analysis on the direct characterizations of the doubly h-BN/graphene/h-BN samples is now added in the Supplementary Information in the revised version of our manuscript. We appreciate the Referee#2 for her/his valuable suggestion, which improves the paper and makes our results much more rigorous.

Comment 2: *“Similar to Hofstadter butterfly, when two lattice with small twist angle form a beating pattern (or Moiré pattern), avoided-crossing happens, gap opens and resistive peaks arises at the carrier density corresponding to the periodicity of the superlattice. The author fig. 1d showed just that, from one set of Moiré at h-BN/graphene interface. This is just basic solid-state physics at play and the gap is a consequence of avoided-crossing of single-electron bands instead of correlation. Zooming-into smaller range (fig.3) reveals peaks that could correspond to higher order Dual Moiré, but the insulating behavior as well as the extracted gap can simply be a result of avoided-crossing as well in absence of correlation, which has been reported multiple times in similar previous studies.”*

Response: We appreciate that the Referee#2 raised her/his concerns about the origins of the pseudogaps experimentally observed in our work. We fully agree that in the case of h-BN/graphene Hofstadter butterfly, the band folding gives rise to the

^[R5] Wang, Zihao, *et al.* Science advances **5**, eaay8897 (2019).

^[R6] McGilly, Leo J., *et al.* Nature Nanotechnology **15**, 580-584 (2020).

avoided-crossing at the mini Brillouin zones at the K-K' points of the host graphene electronic bands, which is the well known so-called 'cloning' of Dirac points.

This is the picture illustrated in Fig.2b-c in the main text of our manuscript. When both twist angles from top and bottom h-BN are zero (i.e., the ideal case, $\theta_t = \theta_b = 0$), the two sets of moiré superlattice from both top and bottom surfaces degenerate into a single one (which is about 14 nm in the ideal case, therefore the carrier doping of full filling, i.e., 4 electrons per moiré unit cell, is $\frac{8}{\sqrt{3}\lambda^2}$), and trivial gap openings from

avoid crossing takes place at electron fillings of $-4, -8, -12 n_0$. This however will not yield any pseudo gaps at partial fillings of $-5, -6, -7 n_0$ observed experimentally, which our results suggest to be a consequence of electron correlation. In this revision, we have included theoretical calculations that deal with such correlation using the Hartree-Fock methods. As already depicted in the response to Comment from Referee#1, we give the short discussions again below for convenience of our Referee#2.

FIG. R9. Calculated correlated gaps occurring at fillings of -5, -6 and -7 n_0 . (a)-(c) illustrate the calculated gap openings from inter-valley coherence, with the amplitude of each IVC order to be 1.42, 1.90, and 1.35 meV for fillings of -5, -6 and -7 n_0 .

There are always two types of gaps induced by the Coulomb interaction. The first type is induced by valley or spin polarization, and the other type originates from

inter-valley coherent interaction. Our calculations reveal that the strength of polarization (valley polarization or spin polarization) is smaller than target band's width, so inter-valley coherent interaction is necessary to open the global gap.

As shown in FIG. R9, it is seen that the inter-valley coherent (IVC) orders are calculated to be existing at all fillings, with the amplitude of each to be 1.42, 1.90, and 1.35 meV for fillings of -5 , -6 and $-7 n_0$. Thanks to these IVC order parameters, insulating behavior can occur at low temperatures. It thus explains why the CIs are still taking place even though the interaction-induced valley or spin polarization is not strong enough to open global gaps.

In the mean time, the doubly aligned moiré superlattice also helps to reduce the band width into an appropriate scale (a few tens of meV of band II, as calculated in Fig. 2c in the main text). Overall, the concurrence of a doubly aligned moiré superlattice, and the existence of IVC orders, is the origin of the observed correlated states at fillings of -5 , -6 , and $-7 n_0$, in our work.

With the above theoretical results, together with the updated data as will be shown in the next responses, it suggests that the fillings of -5 , -6 , $-7 n_0$ are not the same origin as those of -4 , -8 , $-12 n_0$. For the former ones, our experimental and theoretical evidences all point to a consistent origin of electron correlations.

Comment 3: *"In addition to verification of filling factors (see my comment 1), it would strengthen the claim of "correlated" states if the author can demonstrate co-existence of a superconducting dome next to the correlated insulating states (or at least a metallic state developing into a superconducting dome), for example, similar to that of reference [3] in the manuscript "Cao, Y. et al. Unconventional superconductivity in magic-angle graphene superlattices. Nature 556, 43{50 (2018)." The feature between -5 and $-6 n_0$ is still far from zero resistance and the temperature dependence of it can be a consequence of peaks at -5 and $-6 n_0$."*

Response: We fully agree with the referee that sometimes superconductivity comes along with the correlated insulating states in twisted graphene systems. However, latest investigations suggest that those gapped behaviors are not necessarily accompanied with the existence of superconductivity,^[R7, R8, R9] although both the two types of behaviors are consequences of electron correlations.

In our results, down to the lowest temperature that we could obtain, which is ~ 50 mK, no evidence of superconductivity is seen. But this does not harm the existence of

[R7] Stepanov, Petr, *et al.* Nature, **583**, 375-378 (2020).

[R8] Arora, Harpreet Singh, *et al.* Nature, **583**, 379-384 (2020).

[R9] Cao, Yuan, *et al.* Nature, **583**, 215-220 (2020).

correlated insulating states, as examples of correlated gaps can be found in such as twisted double bilayer graphene,^[R10, R11] where superconductivity is absent.

Comment 4: “Lastly, at least data from one more control sample (in addition to sample-4 and 12) should be presented in the manuscript and demonstrate reproducibility, with a full set of magneto transport similar to data from S12 data and plotted side-by-side for direct comparison (figure 3 and 4).”

Response: We thank the Referee#2 for her/his constructive comments.

FIG. R10. A third sample S56 with similar behavior as that of samples S12 and S4. From top to down are field effect curves as a function of temperature measured in different samples. Optical images of each corresponding data are shown on the right side.

^[R10] Burg, G. William, *et al.* Physical Review Letters, **123**, 197702 (2019).

^[R11] Shen, Cheng, *et al.* Nature Physics, **16**, 520-525 (2020).

As we described in the previous responses, due to the existence of inhomogeneities of the moiré superlattice in real space, ideal areas in the fabricated samples could be very local. Sometimes among all electrodes only one pair of contacts can give results in the well-aligned regime. Together with some contact issues from nano-fabrications, sample yields were rather low. Nevertheless, we could successfully reproduce the data from different samples. This is really important to show data from more than one sample, we really appreciate our Referee in regard to make our work more solid.

FIG. R11. Magneto-transport of different samples in the alignment regime. From top to down are recorded from S12, S4, and S56, respectively. Notice that sample S56 was measured in its R_{xy} and absolute value is plotted, and the pattern is a bit disturbed due to electron-hole sign switch.

As shown in FIG. R10, we put side-by-side comparison of three different samples

that all show the same temperature dependences in their field effect curves. The broad satellite peak on the hole side gradually deconvolutes into individual peaks at -4, -5, -6, -7, and -8 n_0 at the ground state. Here, the 3rd control sample S-56 we measured four-probe transverse resistance because of the limit of electrodes (only one pair of transverse electrodes gave results in the well-aligned regime, i.e., very small θ_t and θ_b).

A side-by-side comparison of magneto-transport for these three samples is also shown in FIG. R11, and has been added in the Suppl. Info. in the revised manuscript.

Comment 5: *“Minor detail: Reference [28] and [31] are the same paper that appeared twice as separate references.”*

Response: We feel sorry for the mistake. Indeed [31] appeared to be the same as [28]. We have updated the citations in the revised manuscript, and the two references are merged into a single one. We thank our referee for her/his careful reading.

Response to Reviewer #3:

General Comment: *The authors investigate transport properties of monolayer graphene aligned to both top and bottom encapsulating hexagonal boron nitride (h-BN). They report multiple conductivity minima away from the main Dirac point, which are interpreted by the emergence of higher energy band. In addition, the authors report evidence of correlated insulating states at integer fillings of -5, -6, -7 electrons per moiré unit cell.*

This paper is highly relevant to the topic of moiré physics, which has generated intense interests in the field of condensed matter and material engineering research. It is recently demonstrated that aligning graphene with h-BN generates unexpected physical structures. For example, ferromagnetism emerges from the moiré lattice when ABC trilayer graphene or magic-angle tBLG are aligned with h-BN. Even bernal bilayer graphene, which has been studied extensively in prior efforts, exhibits unexpected ferroelectricity when aligned with h-BN. Based on these recent developments, one could argue that it is of great interest to examine the influence of doubly aligned h-BN/graphene/h-BN alignment.

Indeed, the authors report a few interesting results in their transport measurements. The multiple resistance peaks away from the main Dirac point offers evidence that multiple energy bands are accessible by electrostatic doping. If this is due to the band flattening effect of the double alignment between graphene and h-BN, it could feature strong electron correlation, which will attract future efforts studying this structure. In addition, incipient insulating states are shown to emerge at almost every integer filling of the moiré unit cell. These states are interpreted as correlated insulators stabilized by Coulomb interaction, consistent with flat energy band and strong correlation.

Overall, I find the manuscript to be innovative and well written. It deserves publication in Nature communication, if the authors could address the following comments.

Response: We thank Reviewer #3 for her/his positive comments. We feel glad that the reviewer found our work “*highly relevant to the topic of moiré physics, which has generated intense interests in the field of condensed matter and material engineering research*” and “*to be innovative and well written*”. Based on the reviewer’s comments, we have performed new experiments and analyses that confirm our findings. In following, we provide the point-by-point responses to the reviewer’s comments in detail.

Comment 1: “when graphene is aligned with both top and bottom h-BN, two moiré patterns emerge, between graphene and top (bottom) h-BN. As the twist angle is usually slightly different, these two moiré lattice feature different lattice constant, giving rise to two sets of satellite peaks, one from each moiré, as shown in Ref. [15]. Is the same phenomenon observed here?”

Response: We thank the referee for the valuable comment. Yes, as discussed in Ref. [15], they assumed in their paper that one of the h-BN is perfectly aligned (0 degree) with graphene, while another is slightly twisted with an angle θ , then there will be two sets of satellite peaks one from the 0-degree Graphene/h-BN interface, another from the θ -degree Graphene/h-BN interface. Indeed, some similar phenomenon is observed here in this work.

For example, same phenomenon of such emerged wavelengths of moiré superlattices can be further generalized using a geometric model,^[R] which first assume a rather well-aligned h-BN/graphene interface with a fixed wavelength L_α around 14 nm (Red arrows shown in FIG. R12a), and the other ‘less-aligned’ moiré (L_β , green arrows shown in FIG. R12a) will then have six vectors (blue arrows in FIG. R12a) which are incurred due to the composite moiré super lattice. Experimentally, by varying the $\theta^{\alpha,\beta}$, one can have the solution of six vectors (corresponding to the resulted super moiré superlattice) as a function of $\theta^{\alpha,\beta}$, as shown in FIG. R12b, with the reciprocal vector of each written as (Eq. (2) in the main text):

$$l_\beta = \frac{4\pi}{\sqrt{3}a} \sqrt{\delta^2 + \theta^{\alpha,\beta 2}}$$

FIG. R12. Illustration of geometrical solution of super moiré super lattices formed in a h-BN/graphene/h-BN heterostructure. (a) Assuming a well-aligned (Red vectors) moiré, and rotating another h-BN/graphene moiré in order to find the solutions in agreement with experimentally obtained data, indicated in (b)-(c). Figure (a)-(c) are adapted from Ref. [R1] (Ref. [17] in the main text).

In general, in terms of the above mentioned geometric analysis of the resistive satellite peaks, Ref.[15] has some common aspects as compared to the devices with slight twist in our work.

However, as also discussed in the response to the General Comment from Referee#2, Ref.[15] investigated transport measurements mainly for a device with 0-0-60 degree alignment, where the correlated physics is absent in that regime, since the flat bands are only expected at 0-0-0 alignment as indicated by our theoretical considerations. The physics of correlated states at the very small twisting angle (in the vicinity of 0-0-0 degree alignment) in fillings of -5 , -6 , $-7 n_0$ are the main findings in our work which differ from Ref.[15]. And more theoretical discussions on the origins of these correlated states are given in the following responses.

Comment 2: “related to the last question, how are the twist angles determined in Fig.2e? For example, sample 12 is labeled with 0 and 0.5 degrees. I assume these angles are determined based on the position of respective satellite peaks, and it will be helpful to show these discussions in the main text.”

Response: Continuing with the response to the previous comment from our Referee#3, yes, the twist angles in Fig. 2e are determined using the geometrical model by finding the best $\theta^{\alpha,\beta}$, which yields solutions of L_β that are in agreement with satellite peaks in the field effect curve, with each peak located at electron density of

$$n_\beta = \frac{8}{\sqrt{3}L_\beta^2} \text{ (i.e., full filling of each moiré unit cell area).}$$

FIG. R13. Sample longitudinal resistance plotted in parameter space of B and ν . The n can then be calculated and be related to the corresponding gate voltage. Data plotted from Sample S2. This manner gives well defined n , which is the same as measuring the Hall coefficient in those samples with R_{xy} available.

It is noticed that in order to have a relatively precise determination of the L_β , one has to obtain well defined carrier density n . Here, for the longitudinal resistance measurements, we obtained the n by extracting quantum Hall filling fractions ν at high magnetic field B , since ν is related to n via the formula $\nu = \frac{nh}{Be}$, where h is the Planck constant, and e is the electron charge.

With the calibrated carrier density, we can then plot the field effect curve at zero magnetic field, onto the same scale with calculated $n_\beta = \frac{8}{\sqrt{3}L_\beta^2}$ for all the six solutions using the above mentioned geometrical model. Using Sample S2 as an example, the ‘fixed’ moiré can be regarded as at the doping of the vertical red lines, which is of about 0.5 degree twist. Then, the best way of finding other solutions will be the black solid line with twist angle of 1.53 degree (see figure caption in FIG. R14). Using this empirical method,^[R] we can then find the twist angles of our samples as shown in the main text.

FIG. R14. Finding the twist angle with geometrical model. By finding the best θ , it yields solutions of L_β that are in agreement with satellite peaks in the field effect curve, with each peak located at electron density. In FIG. R14, the black solid circles are the calculated solutions which are the crossing points of the horizontal black solid line and the blue calculated lines. As indicated by black dashed lines, they correspond to the resistive peaks on the experimental field effect curve, with calibrated carrier density using the method in FIG. R13. This figure is also presented in the Supplementary Information referenced as Suppl. Fig. 6

Comment 3: “while CIs are observed at 0/0.5 degree, no CIs at 0.45 degrees. How big is the error bar in determining twist angles?”

Response: We appreciate the great suggestion from the referee, indeed when applying the above empirical equation (i.e., eq. [2] in the main text), one has to select the ‘fixed’ first set of moiré pattern, and to match the rest of moiré wavelengths, which can be of fluctuations.

Even though the CIs are sensitive to twist angles, but as suggested by the Referee, a difference of 0.05 degree should not yield that remarkable change of sample behavior. However, except for AFM scans as described in our response to Comment 1 from Referee#2, this geometrical method is one of the very few in-direct ways of extracting an estimation of twist angles. Due to the small statistics, we regret that we could not give a real error bar at this stage. The uncertainty occurred in the analysis cannot be mitigated, and we are simply reporting what we obtained using Eq.(2), which is only a reference for the possible arrangement of the super moiré lattices. We really appreciate that our Referee#3 pointed this out. Nevertheless, we would be happy to make this response-to-referee letter open for download if the readers would have similar concerns on this issue.

Comment 4: “in Fig.3 and 4, a series of CIs are shown to appear at integer fillings. CIs between filling -4 to -8 are particularly robust compared to other filling. Does this reflect properties that are specific to band II? If so, it should be discussed in the manuscript”

Response: We thank this very constructive suggestion by our referee. This is indeed a very important issue.

As we have addressed in the previous responses, we have performed theoretical analysis to elucidate the specific properties in band II. We write it down below again for the convenience of Referee#3. We have included theoretical calculations that deal with such correlation using the Hartree-Fock methods in band II.

There are always two types of gaps induced by the Coulomb interaction. The first type is induced by valley or spin polarization, and the other type originates from inter-valley coherent interaction. Our calculations reveal that the strength of polarization (valley polarization or spin polarization) is smaller than target band’s width, so inter-valley coherent interaction is necessary to open the global gap. As shown in FIG. R15, it is seen that the inter-valley coherent (IVC) orders are calculated to be existing at all fillings, with the amplitude of each to be 1.42, 1.90, and 1.35 meV for fillings of -5, -6 and -7 n_0 . The IVC orders bring the system into an insulating

behavior at low temperatures.

In the mean time, the doubly aligned moiré superlattice also helps to reduce the band width into an appropriate scale (a few tens of meV of band II, as calculated in Fig. 2c in the main text). While, even in the existence of dual moiré, band I is still too large with a width exceeding 100 meV, and band III may be too much influenced by the hybridization of lower energy bands.

To summarize, the concurrence of a doubly aligned moiré superlattice, and the existence of IVC orders, is the origin of the observed correlated states at fillings of -5, -6, and -7 n_0 , in our work. We have added this part of discussion in the main text, highlighted in blue in Page 7 in the revised manuscript.

FIG. R15. Calculated correlated gaps occurring at fillings of -5, -6 and -7 n_0 . (a)-(c) illustrate the calculated gap openings from inter-valley coherence, with the amplitude of each IVC order to be 1.42, 1.90, and 1.35 meV for fillings of -5, -6 and -7 n_0 .

Comment 5: “Fig.4 tried to argue for an insulating feature along the red dashed line, while the width of the feature is bigger than the separation between the red and white dashed lines. Why is this data set only measured up to 2.5T? A Landau fan measurement at higher field could be more convincing for the point the authors are trying to make here.”

Response: We appreciate the Referee’s comment. The Landau fan measurement at higher magnetic field up to 9 T at 50 mK is given below in FIG. R16, shown also in Supplementary Figure 12. However, we notice that data at the hole side at high magnetic field are very noisy, which cannot be really interpreted to strength up the red dashed line discussed at low fields in the main text. To make the paper more rigorous, we have deleted the discussion of the indication from the red dashed line, which does not affect the major message of correlated states from dual moiré and the IVC states in our system.

FIG. R16. Landau fan diagram of sample S12. Hole side data at high magnetic field are of low quality, which may be due to the contact issue. Therefore, we focus on the low field parts in Fig. 4 in the main text.

Comment 6: “A few more minor points:

(1) the energy gap labeled in Fig.3b appears to be wrong. For example, the slope of the 9.05meV line is smaller than that of the 6.82meV.”

Response: We appreciate the very constructive comments from our Referee#3. Actually, since this question is also raised by Referee#1, we will write the response down below again.

In fact, in many of the systems with thermally activated gaps, the resistance versus temperature (let’s call it R - T curve) can be very different in different

temperature regimes. It is therefore rather critical in choosing the temperature range while extracting the thermal gaps. In the twisted graphene systems, there are a few reports that study the correlated insulating gaps.

However, in our doubly-aligned h-BN/graphene/h-BN system, the temperature dependence of the pseudo-gaps at fillings of -5 , -6 , and -7 n_0 are developed only at very low temperature regime (i.e., below 50 K as shown in Fig. 3a in the main text). Even in this case, the temperature dependences are quite weak. We thus fully agree with the referee that it is very important to re-perform the thermal gap extractions in the lower temperature range rather than the previously used one (50 to 250 K, in the 1st submission), where the results seem to be more coherent.

FIG. R17. Thermal activation gaps at integer fillings. Conductance of the sample at different fillings plotted in a log scale, as a function of $1/T$. Here, the temperature range is set to be 5 to 50 K.

By fitting the thermal activation gap in the temperature range of 5 to 50 K, as shown in FIG. R17, one obtains thermal gaps of about 10.64 meV for the main Dirac point, while those for fillings of -4 and -8 n_0 are estimated to be 5.49 and 2.30 meV, respectively. Notably, for the correlated insulating gaps at fillings of -5 , -6 , and -7 , the thermal gaps are estimated to be even smaller, with each value to be 0.06, 0.47, and 0.22

meV, respectively. The updated figure is now included in the revised manuscript in Fig. 3b, with the descriptions updated in the main text, as well.

Moreover, as shown in FIGR18, we put together the fitted gaps from samples #S4 and #S12 for direct comparison. It is seen that the experimentally observed thermal gaps are in good agreement in these two samples. The figure is also added in the revised Supplementary Information.

FIG. R18. Thermal activation gaps at integer fillings for samples #S4 and #S12. It is seen that the gaps fitted from two samples, within the temperature range of 5 to 50 K, are in good agreement.

Comment 7: “(2) in the second paragraph of the RESULTS section, the perfect-alignment regime is used without definition (it is defined several sentences later. It will be helpful to define the term sooner). The term perfect alignment is also misleading. It sounds like perfect alignment (0 degree), even though it refers to <1degree. Maybe “alignment regime” is more appropriate.”

Response: Indeed, “alignment regime” is more appropriate. We have updated the definition of this term in the main text. First, the term is re-defined in an earlier place in the main text. Second, it is now defined as “well-aligned regime”, which refers to $\theta < 1$ degree, in the revised manuscript in page 3. We thank the Referee#3 for her/his valuable suggestion.

Comment 8: “(3) left column on page 5, typo in “due to possible existing stains...””

Response: Thanks a lot for the careful reading, and we have updated the typo ‘stains’ with ‘strains’ in the revised manuscript.

REVIEWER COMMENTS

Reviewer #1 (Remarks to the Author):

The authors have improved the manuscript considerably. However, there are still a few remaining issues.

- 1) English should be checked. There are several sloppy mistakes. For example, in abstract 'the multiplies' should be 'multiples'; further in the text: 'filed'  'field'; 'Fan'  'fan'; 'super lattice'  'superlattice'; 'an role'  'a role'; 'quanta' is often used for a single 'quantum', quanta is the plural of quantum.
- 2) Fig. 1(e) and Figs. 2,7,12,13 show two curves. The difference between the two curves are not explained. I guess one is for sweep up and the other for sweep down? If so, mention it and distinguish them by adding arrows.
- 3) Figs. R2 and R9 (Figs. 4(c,d,e)): what is shown? Figure caption mentions 'correlated gaps'. Only in the main text the actual meaning of the two curves is given as being the band structure.
- 4) Comment 3 of my report has been poorly addressed. In Figs. R4 and 4(b) the point where there are sign changes in R_{xy} is displaced from $n/n_0 = -6, -4$. What is the reason? Can this be a consequence of mixing of R_{xx} in the result due to some misalignment of the Hall probes? Fig. R3: what is the magnetic field at which R_{xy} is shown?
- 5) The thermal gaps of the CIs (Fig. 3(b)) are compared with the theoretical calculated band width which are found to be much larger. I would expect that one should compare those thermal gaps with the energy gaps between the bands. Therefore, a similar Fig. 2(C) should be presented with the energy gaps instead of band widths.
- 6) The band gaps are obtained from a fitting of the high temperature conductance. I would expect that it should be done in the low temperature range. For high temperature the conductance is also influenced by scattering processes on e.g. acoustic phonons.
- 7) In the discussion section it is mention that the IVC order parameter amplitude is of the order 1-2 meV as seen from Figs. 4(c-e). I do not see this amplitude from those figures. It is not clear to me what the authors mean with 'amplitude'. Are they referring to the band gap between the bands in Figs. 4(c-e)? Please indicate the amplitudes in those figures.
- 8) In my view, there is no clear experimental evidence that the origin of the experimental observations are due to correlated states as a consequence of inter-valley coherence enhanced electron interaction. The IVC is not well discussed and it is inferred from theoretical calculations that are not presented in the present work and are referred to a future publication. At present this conclusion of the relevance of IVC is therefore based on speculation.

Reviewer #2 (Remarks to the Author):

The author addressed some of my previous comments satisfactorily. The new AFM imaging of topology is very convincing. I am convinced that the superlattice is there, but this alone is an incremental progress compared to previous works on hBN/G/hBN superlattices with smaller twist angle. The potential novelty and impact of the paper is the correlated states at the integer fillings of the super Moire cell. My remaining concern is still the rather assertive interpretation of the observed insulating behavior, which in my opinion is still rather inconclusive whether it can or should be attributed to correlated insulating states, even with the new data/narrative/analysis. The added transport data is helpful, but mostly qualitative comparison. While the main argument of the correlation relies on quantitative analysis of gap size and how it can not be explained by single-particle picture alone. More comprehensive measurement and quantitative analysis similar to that performed on previous device is necessary. More details on how the calculation is performed should also be provided. I would recommend reconsideration for publication in Nature Communication if the author can address the following comments to strengthen the claims on

correlated states:

Specifically,

1. The thermal activation gap extracted for $\nu = -5, -6, -7$ is 0.06 meV, 0.47 meV and 0.22 meV. 0.06 meV correspond to less than 1K in temperature, even for $\nu = -6, -7$ the thermal activation gap is still well-below 10K, but data used to extract the activation gap for all three states are between 10k - 20k? How is this physical? And how can one reliably assert gap-size extracted from such analysis suggest valley/spin polarization alone can not account for the value extracted here, and that the inter-valley coherent order has to be involved? The claim of correlated states relies on these quantitative arguments, but the way these quantities (particularly gap-size) are extracted does not seem to be reliable.

2. The Hartree-Fock calculation predicts "IVC order to be 1.42, 1.90, and 1.35 meV for fillings of $\nu = -5, -6$ and $\nu = -7$ ". How well does these numbers match with or quantitatively explain experimental results from device 1. How many fitting/variable parameters are used? Does the data from newly fabricated (and measured) device for $\nu = -5, -6, -7$ peak follow the same trend?

3. Can the author provide quantitative gap size extraction on the new device similar to that performed on the previously-measured device? Does the thermal activation gap match with the previous device? If not, can the difference be explained by the Hartree-fock model with minimum change of material parameters? What material parameters needs to be changes to make it work for the new device as well? Is there a qualitative picture such material parameter can vary due to realistic device variations such as local angle inhomogeneity, atomic strain/reconstruction etc?

4. It is true that correlated states does not necessary accompany superconductivity and their microscopic mechanism can be different. However, nearby superconductivity or zero resistance states, while not necessary, can still be useful in suggesting that many-body coherent physics is at play. A sample like that can be particularly useful especially when gap-size extracted is so small to a point analysis/conclusion built upon it is on somewhat inconsistent and shaky ground.

Reviewer #3 (Remarks to the Author):

In the revised manuscript and supplementary information section, the authors provided thorough characterizations of their device, including AFM measurement of moire lattice during device assembly. In addition, the optimized energy gap measurement appears to be more consistent with the interpretation that the mechanism underlying insulators at integer filling $\nu = 5-7$ is Coulomb correlation. The fact that similar behaviors are observed in more samples is also encouraging. Overall, the authors have adequately addressed the referees' comments. I recommend the manuscript be published in Nature Communication.

Response to Reviewer #1:

The authors have improved the manuscript considerably. However, there are still a few remaining issues.

Response:

We thank Reviewer #1 for her/his positive comments.

We present here answers to her/his questions in the following, and also have modified the manuscript accordingly. Modified parts are highlighted in blue color in the SI or main text in the revised version of our manuscript.

References/figures appearing in this Response-to-Referees file will be re-indexed if any of them are added in the revised main text (and the Suppl. Info.) of our manuscript.

1) English should be checked. There are several sloppy mistakes. For example, in abstract 'the multiplies' should be 'multiples'; further in the text: 'filed'  'field'; 'Fan'  'fan'; 'super lattice'  'superlattice'; 'an role'  'a role'; 'quanta' is often used for a single 'quantum', quanta is the plural of quantum.

Response:

We appreciate the very careful reading of the Reviewer #1.

We have corrected all these typos, and the changes are highlighted by blue color in the revised manuscript.

2) Fig. 1(e) and Figs. 2,7,12,13 show two curves. The difference between the two curves are not explained. I guess one is for sweep up and the other for sweep down? If so, mention it and distinguish them by adding arrows.

Response:

Indeed, the two curves shown are the trace and re-trace curves recorded by sweeping up and down the gate voltages.

As suggested by Reviewer #1, arrows have been added to distinguish the sweeping directions, and the figure caption of Fig. 1(e) has been updated accordingly.

3) Figs. R2 and R9 (Figs. 4(c,d,e)): what is shown? Figure caption mentions 'correlated gaps'. Only in the main text the actual meaning of the two curves is given as being the band structure.

Response:

Yes, in the previous ‘response-to-referee’ file, Figs. R2 and R9 are band structures. When we mentioned ‘correlated gaps’, it actually means the gap between the conduction and valence band, see the light blue region in the figure below (the new Fig. R1 in this response file). And the Reviewer #1 is totally right, the actual meaning of the curves is given in the main text Fig. 4c-e, as being the band structure.

FIG. R1. Calculated energy bands at the filling of $-7 n_0$. The two valleys without Coulomb interactions are plotted in purple, and the Hartree-Fock (HF) energy bands are given in red. The light blue region indicates the calculated correlated gap with inter-valley coherent (IVC) order involved.

To give an example, we plot here the band structure of $-7 n_0$ in Fig. R1. The bare energy bands of two valleys (purple curve without Coulomb interactions) and Hartree-Fock energy bands (red curves), with the Fermi level set to zero in the plot. We can see that if Coulomb interactions are not taken into consideration, the Fermi level will cross the bands and thus the doped system will be metallic (purple curves). When the Coulomb interaction is taken into account, the energy bands will be split by the interactions, and will behave as a correlated insulator whose gap is indicated by light blue region.

We hope the above explanations will fully address the Reviewer’s question.

4) Comment 3 of my report has been poorly addressed. In Figs. R4 and 4(b) the point where there are sign changes in R_{xy} is displaced from $n/n_0 = -6, -4$. What is the reason? Can this be a consequence of mixing of R_{xx} in the result due to some misalignment of the Hall probes? Fig. R3: what is the magnetic field at which R_{xy} is shown?

Response:

We thank the Reviewer#1 for his/her question.

In the previous round of review, we fabricated sample S-56, which was showing sign changes in R_{xy} at $n/n_0 = -6, -4$. Indeed, this can be a consequence of mixing of R_{xx} due to some misalignment of the Hall probes. The magnetic field was close to zero for the R_{xy} shown in the Fig. R3 in the previous ‘response-to-referee’ letter.

FIG. R2. Longitudinal and transversal resistance of Sample S60 at finite magnetic fields. (a) Color map of R_{xx} and R_{xy} for Sample-S60 in the space of charge fillings and magnetic field. (b) Line cuts of the R_{xx} and R_{xy} at several magnetic fields.

During the past weeks, we have taken our efforts in fabricating new samples, among which Sample-S60 seems to exhibit better alignment of the Hall probes. The data obtained in Sample-S60 at 50 mK is given below in Fig. R2 (which is the updated Fig. 4a-b in the main text).

It is seen that color maps of R_{xx} scanned in n/n_0 and magnetic field show resistive peaks at all integer fillings from $-4 n_0$ to $-8 n_0$, and the corresponding R_{xy} map show sign changes at all integer filling from $-4 n_0$ to $-8 n_0$. Line cuts in Fig. R2 give more details of the R_{xx} and R_{xy} at several typical magnetic fields.

This new dataset obtained Sample S-60 has been updated in the main text in the Fig. 4 in the revised manuscript, in substitution of the previous data measured in Sample S-56.

5) The thermal gaps of the CIs (Fig. 3(b)) are compared with the theoretical calculated band width which are found to be much larger. I would expect that one should compare those thermal gaps with the energy gaps between the bands. Therefore, a similar Fig. 2(C) should be presented with the energy gaps instead of band widths.

Response:

We appreciate the constructive comments from Reviewer #1.

Bandwidths in Fig. 2c are calculated using the conventional continuum model without Coulomb interactions. As suggested by the Reviewer #1, to show the twist angle θ_G (with the top and bottom h-BN kept doubly aligned with respect to the middle graphene layer, as illustrated in Fig. 2a in the main text) dependence of energy gap, we performed calculations of the Hartree-Fock energy bands (which take into account the Coulomb interactions) for different θ_G below 0.7 degree.

As shown in Fig. R3, the correlated global gaps extracted from Hartree-Fock calculations is plotted against θ_G . It is seen that increasing θ_G will diminish the correlated gap of each band of $-5 n_0$, $-6 n_0$, and $-7 n_0$, since the Coulomb interactions are weakened in the doped system at larger θ_G . It indicates that, when the two sandwiching h-BN flakes are well aligned, in order to observe these correlated states, one has to ensure a θ_G less than ~ 0.5 degree.

This new analysis has been updated as the Supplementary Figure 21.

We would like to thank the Reviewer #1 for her/his very helpful comments that have made our manuscript of much improved scientific rigour.

FIG. R3. Correlated gaps obtained from Hartree-Fock calculations. The increase of θ_G will diminish the correlated gap of each band of $-5 n_0$, $-6 n_0$, and $-7 n_0$.

6) The band gaps are obtained from a fitting of the high temperature conductance. I would expect that it should be done in the low temperature range. For high temperature the conductance is also influenced by scattering processes on e.g. acoustic phonons.

Response:

We fully agree with the referee that indeed at higher temperature (such as above 100 K), scattering process will significantly affect the conductance in the system.

In our experiment, the band gaps are obtained from the temperature range of ~ 10 to 50 K. We noticed that at temperature lower than 10 K, the conductance tends to saturate, and the fitting of thermal activation will yield meaningless results.

We would like to emphasize that, in the many reported studies on twisted graphene systems, similar temperature ranges are often used for the fitting of thermal activation gaps for such weak resistive peaks (see references listed below). This is because those gaps are not as robust as band insulators, and are often defined as incipient gaps. In transport experiment, they turn out to be rather weak resistive peaks that cannot develop into more insulating ground state, and become saturating at temperatures below a few Kelvin.

Below are some recent examples of thermal activation fittings in twisted graphene or graphene/h-BN related systems:

1) Thermal gaps extracted in the T -range of 100- 200 K, in h-BN/graphene/h-BN moiré systems. [N. R. Finney, *et al.*, “Tunable crystal symmetry in graphene–boron nitride heterostructures with coexisting moiré superlattices”, *Nature Nanotechnology*, **14**, 1029-1034 (2019).]

2) Thermal gaps extracted in the T -range of 40- 200 K, in graphene aligned with h-BN samples. [B. Hunt, *et al.*, “Massive Dirac Fermions and Hofstadter Butterfly in a van der Waals Heterostructure”, *Science*, **340**, 1427 (2013).]

3) Thermal gaps extracted in the T -range of 10- 100 K, in trilayer graphene/h-BN system. [G. Chen, *et al.*, “Evidence of a gate-tunable Mott insulator in a trilayer graphene moiré superlattice”, *Nature Physics*, **15**, 237–241 (2019).]

4) Thermal gaps extracted in the T -range of about 1-20 K, in magic angle bilayer graphene. [X. Lu, *et al.*, “Superconductors, orbital magnets and correlated states in magic-angle bilayer graphene”, *Nature*, **574**, 653–657 (2019).]

7) In the discussion section it is mention that the IVC order parameter amplitude is of the order 1-2 meV as seen from Figs. 4(c-e). I do not see this amplitude from those figures. It is not clear to me what the authors mean with 'amplitude'. Are they referring to the band gap between the bands in Figs. 4(c-e)? Please indicate the amplitudes in those figures.

Response:

We appreciate the comments by Reviewer #1. Indeed, the previous manuscript was not clear enough in this point.

FIG. R4. Amplitude of the calculated IVC orders. (a)-(c) are the distribution of IVC order in the reciprocal space, at the fillings $\nu=-5$, -6 and $-7 n_0$, respectively.

Actually, the ‘amplitude’ of IVC order parameter refers to the strength of

interactions decomposed into the IVC channel. In other words, the IVC order amplitude is defined as $U^* \langle \rho_{IVC} \rangle$, where U refers to the Coulomb interaction, and $\langle \rho_{IVC} \rangle$ refers to the expectation value of the IVC component of the density matrix calculated in the Hartree-Fock ground state.

Here, in Fig. R4, we present the distribution of IVC order in the reciprocal space. The average values (by summing the amplitudes at each k -point and divided by the mini Brillouin zone) are determined to be 1.42 meV, 1.90 meV and 1.35 meV at the fillings $\nu = -5, -6, \text{ and } -7 n_0$ respectively. But their maximum value at some particular k -points can reach 50~90 meV.

It is due to these IVC order parameter, the doped system can behave as correlated insulator.

The above discussion is updated as Supplementary Figure 20, in the revision.

8) In my view, there is no clear experimental evidence that the origin of the experimental observations are due to correlated states as a consequence of inter-valley coherence enhanced electron interaction. The IVC is not well discussed and it is inferred from theoretical calculations that are not presented in the present work and are referred to a future publication. At present this conclusion of the relevance of IVC is therefore based on speculation.

Response:

We thank Reviewer #1 for her/his comments.

In this latest revision, we have now added detailed new discussions including the amplitude of IVC order parameters at each mini band at integer fillings from -5 to -7 n_0 , and the global correlated gap sizes as a function of θ_G .

The discussion of IVC is much improved, and no future publication will be needed.

In addition, we have also fabricated another new batch of samples, with Samples S-60 and S-63 showing similar correlated states behavior, as resistive peaks at integer fillings from -5 n_0 to -7 n_0 are seen at their ground states. The data of the two new samples are shown below in Fig. R5 (also updated in the Supplementary Figure 18 in the revision).

Up to now, we have fabricated about 65 samples, and altogether 5 out of them are exhibiting similar correlated states at integer fillings from -5 to -7 n_0 , which speaks highly the reproducibility of the observed phenomenon (as will also be discussed in

the response to the Reviewer #2). In transport measurements, these correlated states are reflected as resistive peaks at low temperatures, which cannot be explained by conventional band structure without Coulomb interactions. And after careful examinations via Hartree-Fock calculations, we noticed that even fully spin valley polarizations could not match the bandwidth.

Based on the above experimental observations as well as theoretical analysis, we propose an IVC order that can address the overall $e-e$ interaction strength which can be comparable to the bandwidths.

We fully agree with the Reviewer #1 that our explanation of IVC is a hypothesis, which is self-consistent with the experimental observation. Notice that other hypothesis of gap openings from such as CDW-like Mott insulator on triangular lattice, or from strong Coulomb interaction induced valley splitting, can be ruled out, as we are in a weak interaction regime. We have emphasized this point in the revised manuscript, as highlighted in blue, in the first paragraph in Page 8.

In general, the new calculations and discussions are updated in the revised manuscript, which is now of much improved scientific rigour thanks to the valuable suggestions/comments by Reviewer #1.

FIG. R5. Field effect curves as a function of temperature in the new Samples S-60 and S-63. (a) and (c) are the color map of resistance scanned in T and n/n_0 , while (b) and (d) are optical images of each corresponding sample.

Response to Reviewer #2:

General Comment: *The author addressed some of my previous comments satisfactorily. The new AFM imaging of topology is very convincing. I am convinced that the superlattice is there, but this alone is an incremental progress compared to previous works on hBN/G/hBN superlattices with smaller twist angle. The potential novelty and impact of the paper is the correlated states at the integer fillings of the super Moiré cell. My remaining concern is still the rather assertive interpretation of the observed insulating behavior, which in my opinion is still rather inconclusive whether it can or should be attributed to correlated insulating states, even with the new data/narrative/analysis. The added transport data is helpful, but mostly qualitative comparison. While the main argument of the correlation relies on quantitative analysis of gap size and how it can not be explained by single-particle picture alone. More comprehensive measurement and quantitative analysis similar to that performed on previous device is necessary. More details on how the calculation is performed should also be provided. I would recommend reconsideration for publication in Nature Communication if the author can address the following comments to strengthen the claims on correlated states:*

Response:

We appreciate very much the constructive comments by our Reviewer #2.

All his remaining concerns are crucial for improving the quality and scientific rigour of our work. While appreciating his/her positive affirmation on the novelty and impact of our experimental findings, we do have included significant details of calculation, and acquired new dataset with better quality transport measurement.

According to the suggestions by Reviewer #2, a number of new analyses have been updated, which are highlighted in blue in the revised manuscript. And the paper is much improved thanks to these valuable comments.

In the following, we will answer in a point-by-point manner to the questions raised by Reviewer #2, and we wish this new revision will be satisfactory for publication.

1. *The thermal activation gap extracted for -5,-6,-7 is 0.06 meV, 0.47 meV and 0.22 meV. 0.06 meV correspond to less than 1K in temperature, even for -6, -7 the thermal activation gap is still well-below 10K, but data used to extract the activation gap for*

all three states are between 10k - 20k? How is this physical? And how can one reliably assert gap-size extracted from such analysis suggest valley/spin polarization alone cannot account for the value extracted here, and that the inter-valley coherent order has to be involved? The claim of correlated states relies on these quantitative arguments, but the way these quantities (particularly gap-size) are extracted does not seem to be reliable.

Response:

This question is partially overlapping with the Comment 6 by Reviewer #1, and we will discuss in this thread again, with some of the contents similar as those in response to Reviewer #1.

First, we agree with the Reviewer #2 that the thermal gaps extracted (less than 1 meV) are rather small, as compared to the temperature range. As in our experiment, the thermal activation gaps are obtained from the temperature range of ~10 to 50 K.

This is mainly because those ‘gaps’ are not as robust as band insulators, and are often defined as incipient gaps due to relatively weak interaction strength. In transport experiment, they turn out to be weak resistive peaks that cannot develop into more insulating ground state, and become saturating at temperatures below a few Kelvin. At lower temperature range (say, below 10 K), the conductance tends to saturate, and the fitting of thermal activation will yield meaningless results.

We would like to emphasize that, even in the cases of much flatter energy bands, such as magic angle twisted bilayer graphene (with the width of its flat band to be about 10 meV), very small thermal gaps 0.35 meV ($\nu = -2 n_0$), 0.14 meV ($\nu = 1 n_0$), 0.37 meV ($\nu = 2 n_0$), 0.27 meV ($\nu = 3 n_0$) and 0.86 meV ($\nu = 0$; CNP) are extracted from a T -range of about 1 to 20 K, as reported in [X. Lu, *et al.*, “Superconductors, orbital magnets and correlated states in magic-angle bilayer graphene”, *Nature*, **574**, 653–657 (2019).]

In the many other recently reported studies on twisted graphene systems, similar temperature ranges are often used for the fitting of thermal activation gaps for such weak resistive peaks (see references listed below):

1) Thermal gaps extracted in the T -range of 100- 200 K, in h-BN/graphene/h-BN moiré systems. [N. R. Finney, *et al.*, “Tunable crystal symmetry in graphene–boron nitride heterostructures with coexisting moiré superlattices”, *Nature Nanotechnology*,

14, 1029-1034 (2019).]

2) Thermal gaps extracted in the T -range of 40- 200 K, in graphene aligned with h-BN samples. [B. Hunt, *et. al.*, “Massive Dirac Fermions and Hofstadter Butterfly in a van der Waals Heterostructure”, *Science*, **340**, 1427 (2013).]

3) Thermal gaps extracted in the T -range of 10- 100 K, in trilayer graphene/h-BN system. [G. Chen, *et al.*, “Evidence of a gate-tunable Mott insulator in a trilayer graphene moiré superlattice”, *Nature Physics*, **15**, 237–241 (2019).]

Furthermore, we did not claim that the thermal activation gaps are having anything to do with the valley/spin polarization.

In fact, what we would like to emphasize is that the thermal activation gaps are rather small, which is in agreement with other incipient gaps due to electron correlations as reported by the references listed above.

In the mean time, as those resistive peaks develop at integer fillings at -5 to $-7 n_0$, which cannot be explained by conventional non-interacting single particle pictures. Therefore, we performed careful examinations via Hartree-Fock calculations, and we noticed that even fully spin valley polarizations could not match the bandwidth given in Fig. 2c in the main text.

Based on the above experimental observations as well as theoretical analysis, we proposed an IVC order that can address the overall $e-e$ interaction strength which can be comparable to the bandwidths, which is self-consistent with the experimental observation.

In the updated revision, we have included new discussions as highlighted by blue in the revised manuscript, and new details have been added in the Methods section, as well. In brief, the screened Coulomb interaction can be written as $V(q) = U_M q_M \tanh(qd_s)/q$, in which $q_M = 4\pi/\text{sqrt}(3)L_M$ is the length of reciprocal lattice vector, and $U_M = e^2/4\pi\epsilon\epsilon_0 L_M$ represents the characteristic Coulomb interaction. L_M is moiré lattice constant in real space and d_s is the screening length of about 200 Angstrom. It is noticed that, using the above relations, at the $\theta = 0.5$ -degree angle, the amplitude of U_M is only ~ 28 meV, which is roughly the energy scale of the valley and spin splittings if a valley- and/or spin-polarized state were the ground state at partial integer fillings. However, this value is still far lower than the bandwidth of band II (~ 80 meV) shown in Fig. 2 in the main text. Thus, we conjecture that an IVC order is needed in order to open a global gap for such relatively weak Coulomb interaction strength (notice that other hypothesis of gap openings from such as Mott insulator on triangular lattice, or from strong Coulomb interaction induced valley splitting, can be

ruled out, as we are in a weak interaction regime).

And indeed, as shown in the updated Supplementary Figure 20, the IVC interactions can reach a maximum amplitude of 50~90 meV at single k -points. This interaction strength is much stronger than U_M mentioned above, and is comparable to the bandwidth shown in Fig. 2c in the main text. Hence, the correlated global gaps can be opened.

At the end, by calculating the Hartree-Fock band structure with the IVC order involved, we obtain the global correlated gap sizes as a function of twist angle θ_G , shown in Supplementary Figure 21. It is seen that the global gap can reach up to the order of 10-20 meV for the perfectly aligned situation. However, considering the error of twist angle in the dual moiré experiments, the global gap sizes can be much reduced.

In this latest revision, we have now added detailed new discussions including the amplitude of IVC order parameters at each mini band at integer fillings from -5 to -7 n_0 , and the global correlated gap sizes as a function of twist angle θ_G . The IVC discussions are much improved, and the hypothesis of IVC is consistent with our experimental observations.

2. The Hartree-Fock calculation predicts "IVC order to be 1.42, 1.90, and 1.35 meV for fillings of -5, -6 and -7 n_0 ". How well do these numbers match with or quantitatively explain experimental results from device 1. How many fitting/variable parameters are used? Does the data from newly fabricated (and measured) device for $\nu=-5,-6,-7$ peak follow the same trend?

Response:

In our Hartree-Fock calculations, the only variable parameter in the screened Coulomb interaction is the background dielectric constant ϵ , and we take $\epsilon=4$ in the calculations. All other parameters are kept the same with the experimental configurations.

The value of 1.42 meV, 1.90 meV and 1.35 meV at the fillings $\nu=-5, -6$ and $-7 n_0$ are actually the average of ‘amplitude’ of IVC order, which refers to the average strength of IVC order parameter in the ground Hartree-Fock state. The average amplitudes are determined by summing the amplitudes at each k -point and divided by the mini Brillouin zone. Notice that the maximum value of IVC order at single k -points can reach 50~90 meV.

As shown in Fig. R6 (Supplementary Figure 20 in the revision), the distribution of IVC order are plotted in the reciprocal space. But their maximum value at some particular k -points can reach 50~90 meV, and this interaction strength is much

stronger than U_M mentioned in the discussion of the previous question, and is comparable to the bandwidth shown in Fig. 2c in the main text. Hence, the correlated global gaps can be opened.

As shown in the calculated energy bands, when taking IVC order into account at $\theta_G=0.5^\circ$, in Fig. 4c-e in the main text, global gaps openings can be seen as indicated by the light-blue regions, with the value of 11.8, 9.7, and 1.8 meV, for $\nu=-5$, -6, and -7 n_0 , respectively. The calculated correlated gaps are larger than the experimentally extracted thermal activation ones, which may come from the underestimated quantum fluctuations (thus overestimated gaps) in the Hartree-Forck simulations. It is known that the gap itself also strongly depends on experimental details, such as errors in the twist angle, and the exact dielectric constant in the materials. Nevertheless, the general trend of the three correlated gaps is in qualitative agreement with the ones obtained in Supplementary Figure 17.

And, yes, the data from the newly fabricated (and measured) devices (S60, and S63) for $\nu = -5, -6, -7 n_0$ resistive peaks follow the same trend, as will be discussed in more details in the response to the next comment.

FIG. R6. Amplitude of the calculated IVC orders. (a)-(c) are the distribution of IVC order in the reciprocal space, at the fillings of $\nu = -5$, -6 and -7 n_0 , respectively.

3. Can the author provide quantitative gap size extraction on the new device similar to that performed on the previously-measured device? Does the thermal activation gap match with the previous device? If not, can the difference be explained by the Hartree-fock model with minimum change of material parameters? What material parameters need to be changes to make it work for the new device as well? Is there a qualitative picture such material parameter can vary due to realistic device variations such as local angle inhomogeneity, atomic strain/reconstruction etc?

Response:

We thank the Reviewer #2 for her/his comment.

Yes, we can certainly provide thermal activation gap extraction on the new devices similar to that performed on the previously-measured device.

FIG. R7. Comparison of field effect curves as a function of temperature for samples S12, S4, S56, S60, and S63. From top to down are field effect curves as a function of temperature measured in different samples, with the optical images of each corresponding data shown on the right side.

In the new revision, we have fabricated another new batch of samples, with Samples S-60 and S-63 showing similar correlated states behavior, as resistive peaks at integer fillings from $-5 n_0$ to $-7 n_0$ are seen at their ground states. The data of the two new samples are shown below in Fig. R7, and also updated in the Supplementary Figure 18 in the revision.

Up to now, we have fabricated about 65 samples, and altogether 5 out of them are exhibiting similar correlated states at integer fillings from -5 to $-7 n_0$, which speaks highly the reproducibility of the observed phenomenon (as will also be discussed in the response to the Reviewer #2).

It is noticed that, except for Sample S-56, which was measured in R_{xy} (some of the electrodes were broken in S-56 and no available R_{xx} could give the correlated peaks, probably due to spatial inhomogeneity of the moiré superlattice in the device), we could extract thermal activation gaps out of 4 samples.

FIG. R8. Thermal activation gaps at integer fillings for samples S4, S12, S60, and S63. It is seen that the thermal activation gaps fitted from the 4 samples, within the temperature range of 10 to 50 K, are in good agreement.

As shown in Fig. R8 (updated Supplementary Figure 17), the thermal activation gaps fitted from the 4 samples are in good agreement.

On one hand, after examining all the samples, we do notice that to see the correlated states at integer fillings from -5 to $-7 n_0$ in the dual-moiré devices, one has to reduce as much as possible the inhomogeneities, as shown in the AFM scans in Supplementary Figures 10-11. In the mean time, the alignment of the dual moiré is very crucial to observe the correlated states.

FIG. R9. Correlated gaps obtained from Hartree-Fock calculations as a function of θ_G . The increase of θ_G will diminish the correlated gap of each band of $-5 n_0$, $-6 n_0$, and $-7 n_0$. A fixed dielectric constant of $\varepsilon=4.0$ is used in the calculation.

FIG. R10. Correlated gaps obtained from Hartree-Fock calculations as a function of dielectric constant. The increase of dielectric constant will also diminish the correlated gap of each band of $-5 n_0$, $-6 n_0$, and $-7 n_0$. A fixed $\theta_G = 0.5^\circ$ is used in the calculation.

On the other hand, we also performed calculations of the Hartree-Fock energy bands (which take into account the Coulomb interactions) for different θ_G below 0.7 degree. As shown in Fig. R9, the correlated global gaps extracted from Hartree-Fock calculations is plotted against θ_G . It is seen that increasing θ_G will diminish the correlated gap of each band of $-5 n_0$, $-6 n_0$, and $-7 n_0$, since the Coulomb interactions are weakened in the doped system at larger θ_G . It indicates that, when the two sandwiching h-BN flakes are well aligned, in order to observe these correlated states, one has to ensure a θ_G less than ~ 0.5 degree.

In the mean time, it is also found that the global gap sizes drastically diminish with an increased dielectric constant, as shown in Fig. R10.

The above new analyses have been updated as the Supplementary Figure 21-22 in the revision.

4. It is true that correlated states does not necessary accompany superconductivity and their microscopic mechanism can be different. However, nearby superconductivity or zero resistance states, while not necessary, can still be useful in suggesting that many-body coherent physics is at play. A sample like that can be particularly useful especially when gap-size extracted is so small to a point analysis/conclusion built upon it is on somewhat inconsistent and shaky ground.

Response:

We are grateful for the very constructive comments from the referee#2.

First, we appreciate that the Reviewer #2 agrees with us that superconducting features are not strictly necessary to claim the observation of correlated insulating state in our system.

Among the over 60 samples fabricated, we have 5 dual-moiré samples showing the correlated states at integer fillings from -5 to $-7 n_0$. However, none of them exhibit superconductivity in a reachable doping range and to the lowest temperature we could obtain.

Possible explanations for the absence of superconductivity in our samples, even though correlated states signatures are observed, could be that the microscopic mechanism of $e-e$ interaction-induced superconductivity in such as magic angle twisted bilayer graphene are different from that in our case.

We have updated the above discussion in the revised manuscript, highlighted by blue in Page 7. Nevertheless, by adding the newly fabricated samples together with more detailed calculations, we believe that the reported results in our manuscript are

much more consistent.

Again, we sincerely appreciate the constructive comments from Reviewer #2. Thanks to his/her deep insight and great suggestions in both experimental and theoretical details, our manuscript has been improved significantly.

We hope the latest revision of our main text, along with the updated Supplementary Information, will be satisfactory.

Response to Reviewer #3:

General Comment: *In the revised manuscript and supplementary information section, the authors provided thorough characterizations of their device, including AFM measurement of moiré lattice during device assembly. In addition, the optimized energy gap measurement appears to be more consistent with the interpretation that the mechanism underlying insulators at integer filling 5-7 is Coulomb correlation. The fact that similar behaviors are observed in more samples is also encouraging. Overall, the authors have adequately addressed the referees' comments. I recommend the manuscript be published in Nature Communication.*

Response:

We appreciate very much the support of publication from Referee #3.

REVIEWERS' COMMENTS

Reviewer #1 (Remarks to the Author):

The authors have replied convincingly to my referee report. I recommend publication.

Reviewer #2 (Remarks to the Author):

It is understandable that assertive conclusions may not be reached for system of this complicity within one manuscript. The data from the new device at least qualitatively follows the same trend and is generally consistent with the physics picture presented. The argument for absence of superconductivity is plausible and the paper by itself (after previous rounds of improvement) is of high standard without it. In that regards, I feel my remaining concerns have been satisfactorily addressed. I recommend publication in Nature Communication, and I thank the patience and time from the Editors and authors in improving the manuscript.